# FEW-SHOT DOMAIN ADAPTATION FOR END-TO-END COMMUNICATION

Jayaram Raghuram [1], Yijing Zeng [1], Dolores García Martí [2], Rafael Ruiz Ortiz [2], Somesh Jha [1,3],
Joerg Widmer [2], Suman Banerjee [1]

[1] University of Wisconsin - Madison      [2] IMDEA Networks Institute, Madrid      [3] XaiPient
```
{jayaramr, yijingzeng, jha, suman}@cs.wisc.edu
{dolores.garcia, rafael.ruiz, joerg.widmer}@imdea.org
```

## ABSTRACT

The problem of end-to-end learning of a communication system using an autoencoder – consisting of an encoder, channel, and decoder modeled using neural networks – has recently been shown to be an effective approach. A challenge faced in the practical adoption of this learning approach is that under changing channel conditions (e.g. a wireless link), it requires frequent retraining of the autoencoder in order to maintain a low decoding error rate. Since retraining is both time consuming and requires a large number of samples, it becomes impractical when the channel distribution is changing quickly. We propose to address this problem using a fast and sample-efficient (few-shot) domain adaptation method that does not change the encoder and decoder networks. Different from conventional training-time unsupervised or semi-supervised domain adaptation, here we have a trained autoencoder from a source distribution that we want to adapt (at test time) to a target distribution using only a small labeled dataset, and no unlabeled data. We focus on a generative channel model based on the Gaussian mixture density network (MDN), and propose a regularized, parameter-efficient adaptation of the MDN using a set of affine transformations. The learned affine transformations are then used to design an optimal transformation at the decoder input to compensate for the distribution shift, and effectively present to the decoder inputs close to the source distribution. Experiments on many simulated distribution changes common to the wireless setting, and a real mmWave FPGA testbed demonstrate the effectiveness of our method at adaptation using very few target domain samples [1].

## 1 INTRODUCTION

End-to-end (e2e) learning of a communication system using an autoencoder has been recently shown to be a promising approach for designing the next generation of wireless networks (O'Shea & Hoydis, 2017; Dörner et al., 2018; Aoudia & Hoydis, 2019; O'Shea et al., 2019; Ye et al., 2018; Wang et al., 2017). This new paradigm is a viable alternative for optimizing communication in diverse applications, hardware, and environments (Hoydis et al., 2021). It is particularly promising for dense deployments of low-cost transceivers, where there is interference between the devices and hardware imperfections that are difficult to model analytically. The key idea of e2e learning for a communication system is to use an autoencoder architecture to model and learn the transmitter and receiver jointly using neural networks in order to minimize the e2e symbol error rate (SER).

The channel (*i.e.*, propagation medium and transceiver imperfections) can be represented as a stochastic transfer function that transforms its input $\mathbf{z} \in \mathbb{R}^d$ to an output $\mathbf{x} \in \mathbb{R}^d$. It can be regarded as a black-box that is typically non-linear and non-differentiable due to hardware imperfections (*e.g.*, quantization and amplifiers). Since autoencoders are trained using stochastic gradient descent (SGD)-based optimization (O'Shea & Hoydis, 2017), it is challenging to work with a black-box channel that is not differentiable. One approach to address this problem is to use a known mathemat-

---

[1]Code for our work: `https://github.com/jayaram-r/domain-adaptation-autoencoder`

ical model of the channel (*e.g.*, additive Gaussian noise), which would enable the computation of gradients with respect to the autoencoder parameters via backpropagation. However, such standard channel models do not capture well the realistic channel effects as shown in Aoudia & Hoydis (2018). Alternatively, recent works have proposed to learn the channel using deep generative models that approximate $p(\mathbf{x} \mid \mathbf{z})$, the conditional probability density of the channel, using Generative Adversarial Networks (GANs) (O'Shea et al., 2019; Ye et al., 2018), Mixture Density Networks (MDNs) (García Martí et al., 2020), and conditional Variational Autoencoders (VAEs) (Xia et al., 2020). The use of a differentiable generative model of the channel enables SGD-based training of the autoencoder, while also capturing realistic channel effects better than standard models.

Although this e2e optimization with a generative channel model learned from data can improve the physical-layer design for communication systems, in reality, channels often change, requiring collection of a large number of samples and frequent retraining of the channel model and autoencoder. For this reason, *adapting the generative channel model and the autoencoder as often as possible, using only a small number of samples* is required for good communication performance. Prior works have (to be best of our knowledge) not addressed the adaptation problem for autoencoder-based e2e learning, which is crucial for real-time deployment of such a system under frequently-changing channel conditions. In this paper, we study the problem of domain adaptation (DA) of autoencoders using an MDN as the channel model. In contrast to conventional DA, where the target domain has a large unlabeled dataset and sometimes also a small labeled dataset (semi-supervised DA) (Ben-David et al., 2006), here we consider a *few-shot DA* setting where the target domain has *only* a small labeled dataset, and no unlabeled data. This setting applies to our problem since we only get to collect a small number of labeled samples at a time from the changing target domain (here the channel) [2].

Towards addressing this important practical problem, we make the following contributions:

- We propose a parameter- and sample-efficient method for adapting a generative MDN (used for modeling the channel) based on the properties of Gaussian mixtures (§ 3.1 and § 3.2).
- Based on the MDN adaptation, we propose an optimal input-transformation method at the decoder that compensates for changes in the channel distribution, and decreases or maintains the error rate of the autoencoder without any modification to the encoder and decoder networks (§ 3.3).
- Experiments on a mmWave FPGA platform and a number of simulated distribution changes show strong performance improvements for our method. For instance, in the FPGA experiment, our method improves the SER by 69% with only 10 samples per class from the target distribution (§ 4).

**Related Work.** Recent approaches for DA such as DANN (Ganin et al., 2016), based on adversarial learning of a shared representation between the source and target domains (Ganin & Lempitsky, 2015; Ganin et al., 2016; Long et al., 2018; Saito et al., 2018; Zhao et al., 2019; Johansson et al., 2019), have achieved much success on computer vision and natural language processing. Their high-level idea is to adversarially learn a shared feature representation for which inputs from the source and target distributions are nearly indistinguishable to a *domain discriminator* DNN, such that a *label predictor* DNN using this representation and trained using labeled data from only the source domain also generalizes well to the target domain. Adversarial DA methods are not suitable for our problem, which requires fast and frequent test-time DA, because of their high computational and sample complexity and the imbalance in the number of source and target domain samples.

Related frameworks such as transfer learning (Long et al., 2015; 2016), model-agnostic meta-learning (Finn et al., 2017), domain-adaptive few-shot learning (Zhao et al., 2021; Sun et al., 2019), and supervised DA (Motiian et al., 2017a;b) also deal with the problem of adaptation using a small number of samples. Most of them are not applicable to our problem because they primarily address novel classes (with potentially different distributions), and knowledge transfer from existing to novel tasks. Motiian et al. (2017a) is closely related since they also deal with a target domain that only has a small labeled dataset and has the same label space. The *key difference* is that Motiian et al. (2017a) address the training-time few-shot DA problem, while we focus on test-time few-shot DA. Specifically, their adversarial DA method requires both the source and target domain datasets at training time, and can be computationally expensive to retrain for every new batch of target domain data (a key motivation for this work is to avoid frequent retraining).

---

[2] In our problem, labels correspond to the transmitted messages and are essentially obtained for free (see § 3).

## 2 PRIMER ON AUTOENCODER-BASED END-TO-END COMMUNICATION

**Notations.** We denote vectors and matrices with boldface symbols. We define the indicator function $\mathbb{1}(c)$ that takes value 1 (0) when the condition $c$ is true (false). For any integer $n \geq 1$, we define $[n] = \{1, \cdots, n\}$. We denote the one-hot-coded vector with 1 at index $i$ and the rest zeros by $\mathbf{1}_i$. The probability density of a multivariate Gaussian with mean $\boldsymbol{\mu}$ and covariance matrix $\boldsymbol{\Sigma}$ is denoted by $\mathcal{N}(\mathbf{x} \,|\, \boldsymbol{\mu}, \boldsymbol{\Sigma})$. We use the superscripts s and t to denote quantities corresponding to the source and target domain respectively. Table 2 in the Appendix provides a quick reference for the notations.

Following (O'Shea & Hoydis, 2017; Dörner et al., 2018), consider a single-input, single-output (SISO) communication system shown in Fig. 1, consisting of a transmitter (or encoder), channel, and receiver (or decoder). The encoder $\mathbf{E}_{\boldsymbol{\theta}_e}(\cdot)$ is a multi-layer feed-forward neural network (NN) with parameters $\boldsymbol{\theta}_e$, that maps an input message $y \in \mathcal{Y} := \{1, \cdots, m\}$ into an encoded symbol $\mathbf{z} \in \mathbb{R}^d$. The input

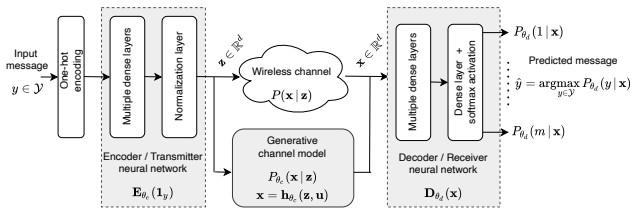

Figure 1: Autoencoder-based end-to-end communication system with a generative channel model.

message $y$ is mapped into a one-hot-coded vector $\mathbf{1}_y$ prior to being processed by the encoder [3]. The message $y$ is equivalent to a class label in machine learning terms, and the encoded symbol $\mathbf{z} = \mathbf{E}_{\boldsymbol{\theta}_e}(\mathbf{1}_y)$ is like a representative vector for the class $y$. We note that the dimension of the encoding $d$ is small (less than 10), and $d = 2$ is typically used to coincide with traditional modulation techniques (O'Shea & Hoydis, 2017; Goldsmith, 2005). The set of distinct encoded symbols $\mathcal{Z} = \{\mathbf{E}_{\boldsymbol{\theta}_e}(\mathbf{1}_1), \cdots, \mathbf{E}_{\boldsymbol{\theta}_e}(\mathbf{1}_m)\}$ is referred to as the *constellation of the autoencoder*.

The symbol $\mathbf{z}$ is transmitted (via the custom modulation learned by the encoder) over a communication channel, represented by an unknown conditional probability density $p(\mathbf{x} \,|\, \mathbf{z})$, and is received at the output of the channel as a noisy, distorted symbol $\mathbf{x} \in \mathbb{R}^d$. The decoder $\mathbf{D}_{\boldsymbol{\theta}_d}(\cdot)$ is also a multi-layer, feed-forward NN with parameters $\boldsymbol{\theta}_d$ that predicts the class-posterior probabilities over the $m$ messages based on the distorted channel output $\mathbf{x}$. The decoder is essentially a classifier whose input-output mapping is defined by $\mathbf{D}_{\boldsymbol{\theta}_d}(\mathbf{x}) := [P_{\boldsymbol{\theta}_d}(1 \,|\, \mathbf{x}), \cdots, P_{\boldsymbol{\theta}_d}(m \,|\, \mathbf{x})]$, where $P_{\boldsymbol{\theta}_d}(y \,|\, \mathbf{x})$ is the predicted probability of class $y$ given $\mathbf{x}$. The class with the highest predicted probability is the decoded message $\widehat{y}(\mathbf{x}) = \mathrm{argmax}_{y \in \mathcal{Y}} \, P_{\boldsymbol{\theta}_d}(y \,|\, \mathbf{x})$. As in standard classification, the performance metric of the autoencoder is the *symbol error rate* (SER), defined as $\mathbb{E}_{(\mathbf{x}, y)}[\mathbb{1}(\widehat{y}(\mathbf{x}) \neq y)]$.

**Generative Channel Model.** In order to learn the encoder and decoder networks using SGD-based optimization, it is necessary to have a differentiable backward path from the decoder to the encoder through the channel. We address this by learning a parametric generative model of the channel $P_{\boldsymbol{\theta}_c}(\mathbf{x} \,|\, \mathbf{z})$ (with parameters $\boldsymbol{\theta}_c$) that closely approximates the true channel conditional density $p(\mathbf{x} \,|\, \mathbf{z})$. There exists a stochastic data generation or sampling function $\mathbf{x} = \mathbf{h}_{\boldsymbol{\theta}_c}(\mathbf{z}, \mathbf{u})$ corresponding to the generative model, where $\mathbf{u}$ captures the random aspects of the channel (*e.g.*, noise and phase offsets; details in Appendix E). In this work, we model the conditional density of the channel using a *set of $m$ Gaussian mixtures*, one per input message (or class) $y \in \mathcal{Y}$:

$$P_{\boldsymbol{\theta}_c}(\mathbf{x} \,|\, \mathbf{z}) = \sum_{i=1}^{k} \pi_i(\mathbf{z}) \, N\big(\mathbf{x} \,|\, \boldsymbol{\mu}_i(\mathbf{z}), \boldsymbol{\Sigma}_i(\mathbf{z})\big), \quad \mathbf{z} \in \{\mathbf{E}_{\boldsymbol{\theta}_e}(\mathbf{1}_1), \cdots, \mathbf{E}_{\boldsymbol{\theta}_e}(\mathbf{1}_m)\}. \tag{1}$$

Here, $k$ is the number of components, $\boldsymbol{\mu}_i(\mathbf{z}) \in \mathbb{R}^d$ is the mean vector, $\boldsymbol{\Sigma}_i(\mathbf{z}) \in \mathbb{R}^{d \times d}$ is the (symmetric, positive-definite) covariance matrix, and $\pi_i(\mathbf{z}) \in [0, 1]$ is the prior probability of component $i$. It is convenient to express the component prior probability in terms of the softmax function as $\pi_i(\mathbf{z}) = e^{\alpha_i(\mathbf{z})} / \sum_{j=1}^{k} e^{\alpha_j(\mathbf{z})}$, $\forall i \in [k]$, where $\alpha_i(\mathbf{z}) \in \mathbb{R}$ are the component prior logits. We define the parameter vector of component $i$ as $\boldsymbol{\phi}_i(\mathbf{z})^T = [\alpha_i(\mathbf{z}), \boldsymbol{\mu}_i(\mathbf{z})^T, \mathrm{vec}(\boldsymbol{\Sigma}_i(\mathbf{z}))^T]$, where $\mathrm{vec}(\cdot)$ is the vector representation of the unique entries of the covariance matrix. We also define the combined parameter vector from all components by $\boldsymbol{\phi}(\mathbf{z})^T = [\boldsymbol{\phi}_1(\mathbf{z})^T, \cdots, \boldsymbol{\phi}_k(\mathbf{z})^T]$.

An MDN can model complex conditional distributions by combining a feed-forward network with a parametric mixture density (Bishop, 1994; 2007). We use the MDN to predict the parameters of the

---

[3]The encoder has a normalization layer that constrains the average power of the symbols (see Appendix D).

Gaussian mixtures $\phi(\mathbf{z})$ as a function of its input symbol $\mathbf{z}$, *i.e.*, $\phi(\mathbf{z}) = \mathbf{M}_{\boldsymbol{\theta}_c}(\mathbf{z})$, where $\boldsymbol{\theta}_c$ are the parameters of the MDN network. The MDN output with all the mixture parameters has dimension $p = k\,(d(d+1)/2 + d + 1)$. While there are competing methods for generative modeling of the channel such as conditional GANs (Ye et al., 2018) and VAEs (Xia et al., 2020), we choose the Gaussian MDN based on i) the strong approximation properties of Gaussian mixtures (Kostantinos, 2000) for learning probability distributions; and ii) the analytical and computational tractability it lends to our domain adaptation formulation. The effectiveness of a Gaussian MDN for wireless channel modeling has also been shown in García Martí et al. (2020).

The input-output function of the autoencoder is given by $\mathbf{f}_{\boldsymbol{\theta}}(\mathbf{1}_y) = \mathbf{D}_{\boldsymbol{\theta}_d}(\mathbf{h}_{\boldsymbol{\theta}_c}(\mathbf{E}_{\boldsymbol{\theta}_e}(\mathbf{1}_y), \mathbf{u}))$, and the goal of autoencoder learning is to minimize the symbol error rate. Since the sampling function $\mathbf{h}_{\boldsymbol{\theta}_c}$ of a Gaussian mixture channel is not directly differentiable, we apply the Gumbel-Softmax reparametrization (Jang et al., 2017) to obtain a differentiable sampling function (details in Appendix E). More background, including the training algorithm of the autoencoder, is in Appendix D.

## 3 PROPOSED METHOD

**Problem Setup.** Let $\mathbf{x}, y, \mathbf{z}$ denote a realization of the channel output, message (class label), and channel input (symbol) distributed according to the joint distribution $p(\mathbf{x}, y, \mathbf{z})$. We first establish the following result about the joint distribution.

**Proposition 1.** *The joint distributions $p(\mathbf{x}, y, \mathbf{z})$ and $p(\mathbf{x}, y)$ can be expressed in the following form:*

$$p(\mathbf{x}, y, \mathbf{z}) = p\big(\mathbf{x}\,|\,\mathbf{E}_{\boldsymbol{\theta}_e}(\mathbf{1}_y)\big)\, p(y)\, \delta(\mathbf{z} - \mathbf{E}_{\boldsymbol{\theta}_e}(\mathbf{1}_y)), \ \ \forall \mathbf{x}, \mathbf{z} \in \mathbb{R}^d,\ y \in \mathcal{Y}$$
$$p(\mathbf{x}, y) = p\big(\mathbf{x}\,|\,\mathbf{E}_{\boldsymbol{\theta}_e}(\mathbf{1}_y)\big)\, p(y), \ \ \forall \mathbf{x} \in \mathbb{R}^d,\ y \in \mathcal{Y}, \tag{2}$$

*where $\delta(\cdot)$ is the Dirac delta (or Impulse) function, and we define $p(\mathbf{x}\,|\,y) := p(\mathbf{x}\,|\,\mathbf{E}_{\boldsymbol{\theta}_e}(\mathbf{1}_y))$ as the conditional distribution of $\mathbf{x}$ given the class $y$.*

The proof is simple and given in Appendix A. Let $\mathcal{D}^s = \{(\mathbf{x}_i^s, y_i^s, \mathbf{z}_i^s),\ i = 1, \cdots, N^s\}$ be a large dataset from a source distribution $p^s(\mathbf{x}, y, \mathbf{z}) = p^s(\mathbf{x}\,|\,y)\, p^s(y)\, \delta(\mathbf{z} - \mathbf{E}_{\boldsymbol{\theta}_e}(\mathbf{1}_y))$. The data collection involves sending multiple copies of each of the $m$ messages through the channel (*e.g.*, over the air from the transmitter to receiver) by using a standard modulation technique (encoding) for $\mathbf{z}$ (*e.g.*, M-QAM (Goldsmith, 2005)), and observing the corresponding channel output $\mathbf{x}$. Different from conventional machine learning, where class labeling is expensive, in this setting the class label is simply the message transmitted, which is *obtained for free* while collecting the data. The MDN channel model and autoencoder are trained on $\mathcal{D}^s$ according to Algorithm 1 (see Appendix D.3).

Due to changes in the channel condition and environmental factors (*e.g.*, moving obstacles), suppose the data distribution changes to $p^t(\mathbf{x}, y, \mathbf{z}) = p^t(\mathbf{x}\,|\,y)\, p^t(y)\, \delta(\mathbf{z} - \mathbf{E}_{\boldsymbol{\theta}_e}(\mathbf{1}_y))$. While the distribution change may cause a drop in the autoencoder's performance, we assume that it is gradual enough that domain adaptation is possible (David et al., 2010) (by domain, here

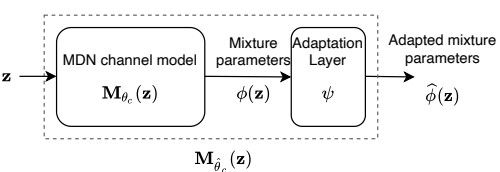

Figure 2: Proposed MDN adaptation method.

we mean the state of the communication channel during the time period when the MDN and autoencoder are trained). As discussed in § 1, the main challenge in this setting is to collect a sufficiently large dataset to retrain the MDN and autoencoder under the distribution shift. Therefore, suppose we collect a small dataset from the target distribution $\mathcal{D}^t = \{(\mathbf{x}_i^t, y_i^t, \mathbf{z}_i^t),\ i = 1, \cdots, N^t\}$, where $N^t \ll N^s$. *Our goal is to design a few-shot domain adaptation for the MDN and autoencoder in order to maintain or improve the symbol error rate.*

**Distribution Change.** Referring to the joint distribution Eq. (2), the class prior $p(y)$ is the prior probability of a message $y$ transmitted through the system. In this work, we make a reasonable practical assumption that this prior probability does not change, *i.e.*, $p^t(y) \approx p^s(y),\ \forall y \in \mathcal{Y}$. However, the class-conditional distribution of channel output $p(\mathbf{x}\,|\,y)$ changes, and therefore the class-posterior distribution $p(y\,|\,\mathbf{x})$ also changes. This is commonly referred to as the *conditional shift* assumption (Zhang et al., 2013) (different from covariate shift (Sugiyama et al., 2007)).

**Overview of the Proposed Method.** Recall from Eqn. (1) that we model the channel distribution $p(\mathbf{x}\,|\,\mathbf{z})$ as a Gaussian mixture $P_{\boldsymbol{\theta}_c}(\mathbf{x}\,|\,\mathbf{z})$, whose parameters are predicted by the MDN, *i.e.*, $\phi(\mathbf{z}) =$

$\mathbf{M}_{\boldsymbol{\theta}_c}(\mathbf{z})$. From Proposition 1, the $m$ *class-conditional distributions* of $\mathbf{x}$ are given by $p(\mathbf{x} \mid y) = p(\mathbf{x} \mid \mathbf{E}_{\boldsymbol{\theta}_e}(\mathbf{1}_y))$, $\forall y \in \mathcal{Y}$. Therefore, in our setting, *adaptation of the class-conditional distributions is equivalent to adaptation of the $m$ Gaussian mixtures* in Eqn. (1. Adaptation of the Gaussian mixtures can be directly accomplished by adapting the MDN (*i.e.*, the parameters $\boldsymbol{\theta}_c$) using the small target-domain dataset $\mathcal{D}^t$. Our proposed adaptation of the autoencoder consists of two *key steps*:

1. A light-weight, parameter-efficient adaptation of the MDN using the small target dataset $\mathcal{D}^t$.
2. An *efficient feature transformation* at the input of the decoder (based on the MDN adaptation) that compensates for changes in the class-conditional distributions.

Our method requires adaptation of *only the MDN (channel model)*, while the encoder and decoder networks ($\boldsymbol{\theta}_e$ and $\boldsymbol{\theta}_d$) remain unchanged, making it amenable to fast and frequent adaptation that requires collecting only a small target dataset each time (few-shot setting).

## 3.1 MDN CHANNEL MODEL ADAPTATION

Our goal is to adapt the $m$ Gaussian mixtures in Eqn (1) that model the source class-conditional distributions. Suppose the $m$ adapted Gaussian mixtures corresponding to the (unknown) target class-conditional distributions are

$$P_{\widehat{\boldsymbol{\theta}}_c}(\mathbf{x} \mid \mathbf{z}) = \sum_{i=1}^{k} \widehat{\pi}_i(\mathbf{z}) N\big(\mathbf{x} \mid \widehat{\boldsymbol{\mu}}_i(\mathbf{z}), \widehat{\boldsymbol{\Sigma}}_i(\mathbf{z})\big), \quad \mathbf{z} \in \{\mathbf{E}_{\boldsymbol{\theta}_e}(\mathbf{1}_1), \cdots, \mathbf{E}_{\boldsymbol{\theta}_e}(\mathbf{1}_m)\}, \quad (3)$$

where $\widehat{\boldsymbol{\theta}}_c$ are parameters of the adapted (target) MDN, and the component means, covariances, and prior probabilities with a hat notation are defined as in § 2. The adapted MDN predicts all the parameters of the target Gaussian mixture as $\widehat{\boldsymbol{\phi}}(\mathbf{z}) = \mathbf{M}_{\widehat{\boldsymbol{\theta}}_c}(\mathbf{z})$ as shown in Fig. 2, where $\widehat{\boldsymbol{\phi}}(\mathbf{z})$ is defined in the same way as $\boldsymbol{\phi}(\mathbf{z})$. Instead of naively fine-tuning all the MDN parameters $\boldsymbol{\theta}_c$, or even just the final fully-connected layer [4], we propose a parameter-efficient adaptation of the MDN based on the affine-transformation property of the Gaussian distribution, *i.e.*, one can transform between any two multivariate Gaussians through a general affine transformation. First, we state some *basic assumptions* required to make the proposed adaptation tractable.

**A1)** The source and target Gaussian mixtures per class have the same number of components $k$.
**A2)** The source and target Gaussian mixtures (from each class) have a one-to-one correspondence between their components.

Assumption A1 is made in order to not have to change the architecture of the MDN during adaptation due to adding or removing of components. Both assumptions A1 and A2 [5] make it tractable to find the closed-form expression for a simplified KL-divergence between the source and target Gaussian mixtures per class (see Proposition 2).

**Parameter Transformations.** As shown in Appendix B.2, the transformations between the source and target Gaussian mixture parameters, for any symbol $\mathbf{z} \in \mathcal{Z}$ and component $i \in [k]$, are given by

$$\widehat{\boldsymbol{\mu}}_i(\mathbf{z}) = \mathbf{A}_i \, \boldsymbol{\mu}_i(\mathbf{z}) + \mathbf{b}_i, \quad \widehat{\boldsymbol{\Sigma}}_i(\mathbf{z}) = \mathbf{C}_i \, \boldsymbol{\Sigma}_i(\mathbf{z}) \, \mathbf{C}_i^T, \quad \text{and} \quad \widehat{\alpha}_i(\mathbf{z}) = \beta_i \, \alpha_i(\mathbf{z}) + \gamma_i. \quad (4)$$

The affine transformation parameters $\mathbf{A}_i \in \mathbb{R}^{d \times d}$ and $\mathbf{b}_i \in \mathbb{R}^d$ transform the means, $\mathbf{C}_i \in \mathbb{R}^{d \times d}$ transforms the covariance matrix, and $\beta_i, \gamma_i \in \mathbb{R}$ transform the prior logits. The vector of all *adaptation parameters* to be optimized is defined by $\boldsymbol{\psi}^T = [\boldsymbol{\psi}_1^T, \cdots, \boldsymbol{\psi}_k^T]$, where $\boldsymbol{\psi}_i$ contains all the affine-transformation parameters from component $i$. The number of adaptation parameters is given by $k\,(2\,d^2 + d + 2)$. This is typically much smaller than the number of MDN parameters (weights and biases from all layers), even if we consider only the final fully-connected layer for fine-tuning (see Table 1). In Fig. 2, the adaptation layer mapping $\boldsymbol{\phi}(\mathbf{z})$ to $\widehat{\boldsymbol{\phi}}(\mathbf{z})$ basically implements the parameter transformations defined in Eqn. (4). We observe that the affine-transformation parameters are not dependent on the symbol $\mathbf{z}$ (or the class), which is a constraint we impose in order to keep the number of adaptation parameters small. This is also consistent with the MDN parameters $\boldsymbol{\theta}_c$ being independent of the symbol $\mathbf{z}$. Allowing the affine transformations to depend of $\mathbf{z}$ would provide more flexibility, but at the same time require more target domain data for successful adaptation.

---

[4]We show in our experiments that both the fine-tuning approaches fail to adapt well.
[5]We perform ablation experiments (Appendix C.4) that evaluate our method under random Gaussian mixtures with mismatched components. We find that our method is robust even when these assumptions are violated.

**Proposition 2.** *Given $m$ Gaussian mixtures from the source domain and $m$ Gaussian mixtures from the target domain (one each per class), which satisfy Assumptions A1 and A2, the KL-divergence between $P_{\boldsymbol{\theta}_c}(\mathbf{x}, K \mid \mathbf{z})$ and $P_{\widehat{\boldsymbol{\theta}}_c}(\mathbf{x}, K \mid \mathbf{z})$ can be computed in closed-form, and is given by:*

$$\overline{D}_{\boldsymbol{\psi}}(P_{\boldsymbol{\theta}_c}, P_{\widehat{\boldsymbol{\theta}}_c}) \;=\; \mathbb{E}_{P_{\boldsymbol{\theta}_c}}\left[\log \frac{P_{\boldsymbol{\theta}_c}(\mathbf{x}, K \mid \mathbf{z})}{P_{\widehat{\boldsymbol{\theta}}_c}(\mathbf{x}, K \mid \mathbf{z})}\right] \;=\; \sum_{\mathbf{z} \in \mathcal{Z}} p(\mathbf{z}) \sum_{i=1}^{k} \pi_i(\mathbf{z}) \, \log \frac{\pi_i(\mathbf{z})}{\widehat{\pi}_i(\mathbf{z})}$$

$$+ \; \sum_{\mathbf{z} \in \mathcal{Z}} p(\mathbf{z}) \sum_{i=1}^{k} \pi_i(\mathbf{z}) \, D_{\mathrm{KL}}\Big(N\big(\,\cdot\,\mid \boldsymbol{\mu}_i(\mathbf{z}), \boldsymbol{\Sigma}_i(\mathbf{z})\big), \, N\big(\,\cdot\,\mid \widehat{\boldsymbol{\mu}}_i(\mathbf{z}), \widehat{\boldsymbol{\Sigma}}_i(\mathbf{z})\big)\Big), \tag{5}$$

*where $K$ is the mixture component random variable. The first term is the KL-divergence between the component prior probabilities, which simplifies into a function of the parameters $[\beta_1, \gamma_1, \cdots, \beta_k, \gamma_k]$. The second term involves the KL-divergence between two multivariate Gaussians (a standard result), which also simplifies into a function of $\boldsymbol{\psi}$.*

The proof and the final expression for the KL-divergence as a function of $\boldsymbol{\psi}$ are given in Appendix A.1. The symbol priors $\{p(\mathbf{z}), \; \mathbf{z} \in \mathcal{Z}\}$ are estimated using the class proportions from the source dataset $\mathcal{D}^s$. We note that this result is different from the KL-divergence between two arbitrary Gaussian mixtures, for which there is no closed-form expression (Hershey & Olsen, 2007).

## 3.2 Regularized Adaptation Objective

From the above analysis, we can formulate the MDN adaptation as the equivalent problem of finding the optimal set of affine transformations (one per-component) mapping the source to the target Gaussian mixtures. To reduce the possibility of the adaptation finding bad solutions due to the small-sample setting, we introduce a regularization term based on the KL-divergence (defined earlier), which constrains the distribution shift produced by the affine transformations. We consider two scenarios for adaptation: 1)

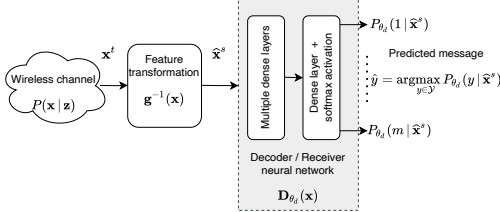

Figure 3: Proposed decoder adaptation using feature transformations.

*Generative adaptation* of the MDN in isolation and 2) *Discriminative adaptation* of the MDN as part of the autoencoder. In the first case, the goal of adaptation is to find a good generative model for the target channel distribution, while in the second case the goal is to improve the classification accuracy of the autoencoder on the target distribution. We focus on the discriminative adaptation here, and present the very similar generative adaptation in Appendix B.3.

Since the goal of adaptation is to improving the decoder's accuracy in recovering the transmitted symbol $\mathbf{z}$ from the channel output $\mathbf{x}$, we use the (negative) symbol posterior log-likelihood (PLL) as the first data-dependent term of the adaptation objective. The second term is the simplified KL-divergence between the source and target Gaussian mixtures, which does *not* depend on the data.

$$J_{\mathrm{PLL}}(\boldsymbol{\psi}\,;\lambda) \;=\; \frac{-1}{N^t} \sum_{n=1}^{N^t} \log P_{\widehat{\boldsymbol{\theta}}_c}(\mathbf{z}_n^t \mid \mathbf{x}_n^t) \;+\; \lambda\,\overline{D}_{\boldsymbol{\psi}}(P_{\boldsymbol{\theta}_c}, P_{\widehat{\boldsymbol{\theta}}_c}). \tag{6}$$

The symbol posterior $P_{\widehat{\boldsymbol{\theta}}_c}(\mathbf{z} \mid \mathbf{x})$ is computed from the conditional $P_{\widehat{\boldsymbol{\theta}}_c}(\mathbf{x} \mid \mathbf{z})$ and the symbol priors $\{p(\mathbf{z}), \; \mathbf{z} \in \mathcal{Z}\}$ using Bayes rule. We observe that the adaptation objective is a smooth and nonconvex function of $\boldsymbol{\psi}$. Also, computation of the objective and its gradient (w.r.t $\boldsymbol{\psi}$) are inexpensive operations since **i)** they *do not require forward and back-propagation* through the layers of the MDN and **ii)** both $N^t$ and the dimension of $\boldsymbol{\psi}$ are small. Therefore, we use the BFGS Quasi-Newton method (Nocedal & Wright, 2006) for minimization, instead of SGD-based large-scale optimization (*e.g.*, Adam). The regularization constant $\lambda$ is a hyper-parameter of the proposed method, and we *propose a validation metric* in Appendix B.4) to set its value automatically.

## 3.3 Decoder Adaptation Using Feature Transformations

We propose a computationally-efficient feature transformation $\mathbf{g}^{-1} : \mathbb{R}^d \mapsto \mathbb{R}^d$ at the decoder such that the transformed inputs $\widehat{\mathbf{x}}^s \;=\; \mathbf{g}^{-1}(\mathbf{x}^t)$ are closely aligned to the source distribution on which

the decoder was trained (see Fig. 3). This is based on the optimal affine-transformations $\boldsymbol{\psi}$ of the MDN found by minimizing Eqn. (6). This method does not require any change to the trained encoder and decoder networks, making it well suited for the few-shot DA setting.

Consider a test input $\mathbf{x}^t$ at the decoder from the target-domain marginal distribution $p^t(\mathbf{x}) = \sum_{\mathbf{z} \in \mathcal{Z}} p(\mathbf{z}) \sum_{i=1}^{k} \widehat{\pi}_i(\mathbf{z}) \, N\big(\mathbf{x} \,|\, \widehat{\boldsymbol{\mu}}_i(\mathbf{z}), \widehat{\boldsymbol{\Sigma}}_i(\mathbf{z})\big)$. As shown in Appendix B.2, conditioned on a given symbol $\mathbf{z} \in \mathcal{Z}$ and component $i \in [k]$, the affine transformation that maps from the target Gaussian distribution $\mathbf{x}^t \,|\, \mathbf{z}, i \ \sim \ N(\mathbf{x} \,|\, \widehat{\boldsymbol{\mu}}_i(\mathbf{z}), \widehat{\boldsymbol{\Sigma}}_i(\mathbf{z}))$ to the source Gaussian distribution $\mathbf{x}^s \,|\, \mathbf{z}, i \sim N(\mathbf{x} \,|\, \boldsymbol{\mu}_i(\mathbf{z}), \boldsymbol{\Sigma}_i(\mathbf{z}))$ is given by

$$\widehat{\mathbf{x}}^s \;=\; \mathbf{g}_{\mathbf{z}i}^{-1}(\mathbf{x}^t) \;:=\; \mathbf{C}_i^{-1}\,(\mathbf{x}^t \,-\, \mathbf{A}_i\,\boldsymbol{\mu}_i(\mathbf{z}) \,-\, \mathbf{b}_i) \,+\, \boldsymbol{\mu}_i(\mathbf{z}). \tag{7}$$

However, this transformation requires knowledge of both the transmitted symbol $\mathbf{z}$ and the mixture component $i$, which are not observed at the decoder (the decoder only observes the channel output $\mathbf{x}^t$). We address this by taking the expected affine transformation from target to source, where the expectation is with respect to the joint posterior over the symbol $\mathbf{z}$ and component $i$, given the channel output $\mathbf{x}^t$. This posterior distribution based on the target Gaussian mixture is:

$$P_{\widehat{\boldsymbol{\theta}}_c}(\mathbf{z}, i \,|\, \mathbf{x}^t) \;=\; \frac{p(\mathbf{z})\,\widehat{\pi}_i(\mathbf{z})\,N\big(\mathbf{x}^t \,|\, \widehat{\boldsymbol{\mu}}_i(\mathbf{z}), \widehat{\boldsymbol{\Sigma}}_i(\mathbf{z})\big)}{\sum_{\mathbf{z}'} \sum_j p(\mathbf{z}')\,\widehat{\pi}_j(\mathbf{z}')\,N\big(\mathbf{x}^t \,|\, \widehat{\boldsymbol{\mu}}_j(\mathbf{z}'), \widehat{\boldsymbol{\Sigma}}_j(\mathbf{z}')\big)}.$$

The expected inverse-affine feature transformation at the decoder is then defined as

$$\mathbf{g}^{-1}(\mathbf{x}^t) \;:=\; \mathbb{E}_{P_{\widehat{\boldsymbol{\theta}}_c}(\mathbf{z}, i \,|\, \mathbf{x})}\big[\mathbf{g}_{\mathbf{z}i}^{-1}(\mathbf{x}^t) \,|\, \mathbf{x}^t\big] \;=\; \sum_{\mathbf{z} \in \mathcal{Z}} \sum_{i \in [k]} P_{\widehat{\boldsymbol{\theta}}_c}(\mathbf{z}, i \,|\, \mathbf{x}^t)\,\mathbf{g}_{\mathbf{z}i}^{-1}(\mathbf{x}^t). \tag{8}$$

We show that this conditional expectation is the *optimal transformation* from the standpoint of mean-squared-error estimation (Kay, 1993) in Appendix A.2. The adapted decoder based on this feature transformation is illustrated in Fig. 3 and defined as $\widehat{\mathbf{D}}_{\boldsymbol{\theta}_d}(\mathbf{x}^t \,;\, \boldsymbol{\psi}) \;:=\; \mathbf{D}_{\boldsymbol{\theta}_d}(\mathbf{g}^{-1}(\mathbf{x}^t))$. For small to moderate number of symbols $m$ and number of components $k$, this transformation is computationally efficient and easy to implement at the receiver of a communication system. A discussion of the computational complexity of the proposed method is given in Appendix B.5.

## 4 EXPERIMENTS

We perform experiments to evaluate the proposed adaptation method for the MDN and autoencoder. Our main findings are summarized as follows: **1)** the proposed method adapts well to changes in the channel distribution using only a few samples per class, often leading to strong improvement over the baselines; **2)** our method performs well under multiple simulated distribution changes, and notably on our mmWave FPGA experiments; **3)** Extensive ablation studies show that the proposed KL-divergence based regularization and the validation metric for setting $\lambda$ are effective.

**Setup.** We implemented the MDN, autoencoder networks, and the adaptation methods in Python using TensorFlow (Abadi et al., 2015) and TensorFlow Probability. We used the following setting in our experiments. The size of the message set $m$ is fixed to 16, corresponding to 4 bits. The dimension of the encoding (output of the encoder) $d$ is set to 2, and the number of mixture components $k$ is set to 5. More details on the experimental setup, neural network architecture, and the hyper-parameters are given in Appendix C.1.

**Baseline Methods.** We compare the performance of our method with the following baselines: **1) No adaptation**, which is the MDN and autoencoder from the source domain without adaptation. **2) Retrained MDN and autoencoder**, which is like an "oracle method" that has access to a large dataset from the target domain. **3) Finetune** - where the method optimizes all the MDN parameters for 200 epochs and optimizes the decoder for 20 epochs [6]. **4) Finetune last** - which follows the same approach as "Finetune", but only optimizes the last layer of MDN (all the layers of the decoder are however optimized). We note that traditional domain adaptation methods are not suitable for this problem because it requires adaptation of both the MDN (generative model) and the decoder.

**Datasets.** The simulated channel variations are based on models commonly used for wireless communication, specifically: i) Additive white Gaussian noise (AWGN), ii) Ricean fading, and iii)

---

[6]We found no significant gains with larger number of epochs in this case.

Uniform or flat fading (Goldsmith, 2005). Details on these channel models and calculation of the their signal-to-noise ratio (SNR) are provided in Appendix F. We also created simulated distribution changes using random, class-conditional Gaussian mixtures for both the source and target channels (we also include random phase shifts). The parameters of the source and target Gaussian mixtures are generated in a random but controlled manner as detailed in Appendix C.3. We also evaluate the performance of the adaptation methods on real over-the-air wireless experiments. We use a recent high-performance mmWave testbed (Lacruz et al., 2021), featuring a high-end FPGA board with 2 GHz bandwidth per-channel and 60 GHz SIVERS antennas (SIVERSIMA, 2020). We introduce distribution changes via (In-phase and Quadrature-phase) IQ imbalance-based distortions to the symbol constellation, and gradually increase the level of imbalance to the system [7]. More details on the FPGA experimental setup are given in Appendix C.2.

**Evaluation Protocol.** Due to the space limit, we provide details of the evaluation protocol such as train, adaptation, and test sample sizes, and the number of random trials used to get averaged performance in Appendix C.1. We report the symbol error rate (SER) on a large held-out test dataset (from the target domain) as a function of the number of target-domain samples per class. The only hyper-parameter $\lambda$ of our method is set automatically using the validation metric proposed in B.4.

## 4.1 AUTOENCODER ADAPTATION ON SIMULATED DISTRIBUTION CHANGES

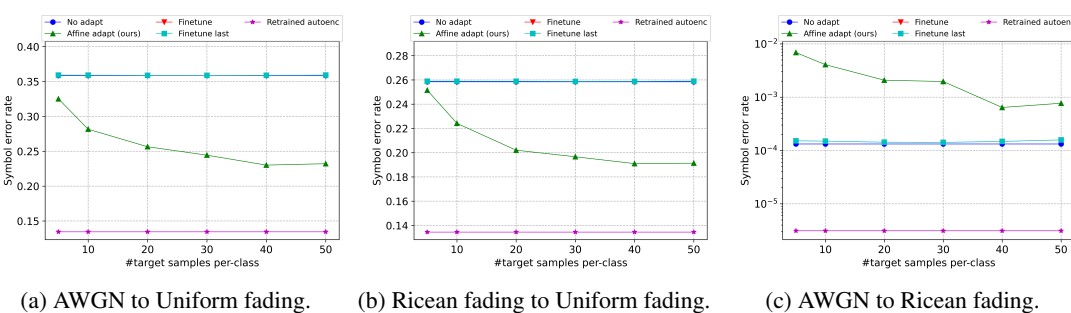

(a) AWGN to Uniform fading.    (b) Ricean fading to Uniform fading.    (c) AWGN to Ricean fading.

Figure 4: Autoencoder adaptation on distribution shifts based on standard channel models.

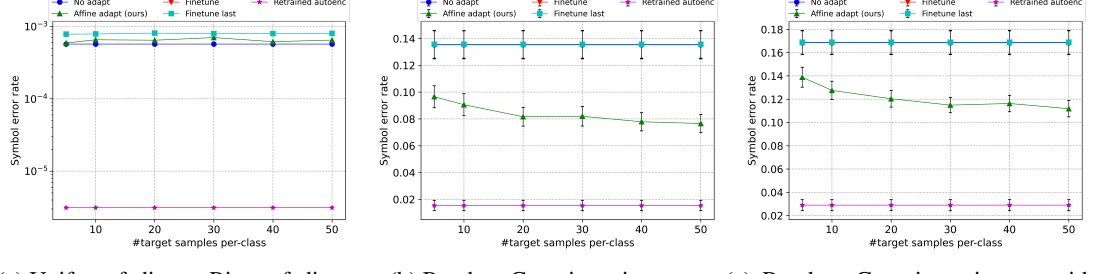

(a) Uniform fading to Ricean fading.    (b) Random Gaussian mixtures.    (c) Random Gaussian mixtures with random phase shifts.

Figure 5: Autoencoder adaptation on distribution shifts based on standard channel models and random Gaussian mixtures. In figure (c), the target domain additionally includes random phase shifts.

The adaptation results under simulated distributions changes are given in Figs. 4 and 5, with the symbol error rates plotted as a function of the number of target samples per class. In Fig. 4, we consider standard channel distributions such as AWGN, Ricean fading, and Uniform fading. In Fig. 5, we consider random Gaussian mixtures for both the source and the target distributions. We observe that the proposed adaptation leads to a strong improvement in SER in all cases, except in the case of AWGN to Ricean fading (Fig. 4. c). We provide some insights on the failure of our method in this case in Appendix C.5. Note that the methods "No adapt" and "Retrained autoenc" have the same SER for all target sample sizes (*i.e.*, a horizontal line). We find both the finetuning baselines to

---

[7]IQ imbalance is a common issue in RF communication that introduces distortions to the final constellation.

have very similar SER in all cases, and there is not much improvement compared to no adaptation. This suggests that our approach of constraining the number of adaptation parameters and using the KL-divergence regularization are effective in the few-shot DA setting (see Table 1).

## 4.2 Autoencoder Adaptation on FPGA Experiments

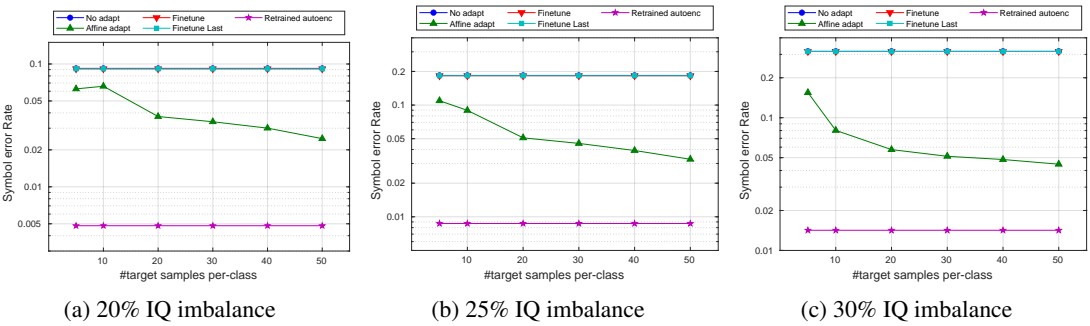

(a) 20% IQ imbalance     (b) 25% IQ imbalance     (c) 30% IQ imbalance

Figure 6: Autoencoder adaptation on the FPGA experiments with distribution change based on IQ imbalance. A higher IQ imbalance results in more distribution change.

For this experiment, different levels of distribution change are introduced by varying the IQ imbalance over 20%, 25%, and 30% (higher IQ imbalance corresponds to larger distribution change). From Fig. 6, we observe that the proposed method achieves significant reduction in error rate compared to the (non-oracle) baselines. The relative improvement in SER over the baselines is more pronounced under higher IQ imbalance. For instance, at 30% IQ imbalance, our method achieves a relative SER improvement of around 69% over the fine-tuning baselines using only 10 samples per-class.

## 4.3 Additional Experiments

We have performed a number of additional experiments including ablation studies, which are reported in Appendix C.4 through C.6. They include: **1)** evaluating the proposed validation metric for automatically setting the hyper-parameter $\lambda$; **2)** evaluating the importance of the KL-divergence regularization in the adaptation objective; **3)** performance of our method when the source and target Gaussian mixtures have a mismatch in the components (addressing Assumptions A1 and A2); **4)** performance of our method when there is no distribution shift; and **5)** performance of the generative adaptation of the MDN channel. To summarize the observations, we found the validation metric to be effective at setting the value of $\lambda$, and that our method has good performance even when Assumptions A1 and A2 are violated, or when there is no distribution shift. The generative MDN adaptation leads to increased log-likelihoods with as low as 2 samples per class.

Table 1: Number of parameters being optimized by the MDN adaptation methods.

| Adaptation method | # parameters | # parameters (specific) |
|---|---|---|
| Finetune | $n_h \, (n_h + d + 2)$ $+ \, k \, (2 \, d + 1) \, (n_h + 1)$ | 12925 |
| Finetune-last-layer | $k \, (2 \, d + 1) \, (n_h + 1)$ | 2525 |
| Proposed | $k \, (d^2 + 2 \, d + 2)$ | 50 |

## 5 Conclusions

In this work, we explore one of the first approaches for domain adaptation of autoencoder based e2e communication in the few-shot setting. We first propose a light-weight and parameter-efficient method for adapting a Gaussian MDN with a very small number of samples from the target distribution. Based on the MDN adaptation, we propose an optimal input transformation method at the decoder that attempts to closely align the target domain inputs to the source domain. We demonstrate the effectiveness of the proposed methods through extensive experiments on both simulated channels and a mmWave FPGA testbed. A discussion of limitations and future directions is given in Appendix B.6.

ACKNOWLEDGMENTS

Banerjee, Raghuram, and Zeng were supported in part through the following grants — US National Science Foundation's CNS-2112562, CNS-2107060, CNS-2003129, CNS-1838733, and CNS-1647152, and the US Department of Commerce's 70NANB21H043. Somesh Jha was partially supported by the DARPA-GARD problem under agreement number 885000. The authors from IMDEA Networks were sponsored by the Spanish Ministry of Economic Affairs and Digital Transformation under European Union NextGeneration-EU projects TSI-063000-2021-59 RISC-6G and TSI-063000-2021-63 MAP-6G, and by the Regional Government of Madrid and the European Union through the European Regional Development Fund (ERDF) project REACT-CONTACT-CM-23479.

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

# Appendix

Table 2: Commonly used notations

| Notation | Description |
|---|---|
| $y \in \mathcal{Y} := \{1, \cdots, m\}$ | Input message or class label. Usually $m = 2^b$, where $b$ is the number of bits. |
| $\mathbf{1}_y,\ y \in \mathcal{Y}$ | One-hot-coded representation of a label (message) $y$, with 1 at position $y$ and zeros elsewhere. |
| $\mathbf{z} \in \mathcal{Z} \subset \mathbb{R}^d$ with $|\mathcal{Z}| = m$ | Encoded representation or symbol vector corresponding to an input message. |
| $\mathbf{x} \in \mathbb{R}^d$ | Channel output that is the feature vector to be classified by the decoder. |
| $\mathbf{E}_{\boldsymbol{\theta}_e}(\mathbf{1}_y)$ | Encoder NN with parameters $\boldsymbol{\theta}_e$ mapping a one-hot-coded message to a symbol vector in $\mathbb{R}^d$. |
| $\mathbf{D}_{\boldsymbol{\theta}_d}(\mathbf{x}) = [P_{\boldsymbol{\theta}_d}(1 \,\|\, \mathbf{x}), \cdots, P_{\boldsymbol{\theta}_d}(m \,\|\, \mathbf{x})]$ | Decoder NN with parameters $\boldsymbol{\theta}_d$ mapping the channel output into probabilities over the message set. |
| $\widehat{y}(\mathbf{x}) = \mathrm{argmax}_{y \in \mathcal{Y}}\ P_{\boldsymbol{\theta}_d}(y \,\|\, \mathbf{x})$ | Class (message) prediction of the decoder. |
| $P_{\boldsymbol{\theta}_c}(\mathbf{x} \,\|\, \mathbf{z})$ | Conditional density (generative) model of the channel with parameters $\boldsymbol{\theta}_c$. |
| $\boldsymbol{\phi}(\mathbf{z}) = \mathbf{M}_{\boldsymbol{\theta}_c}(\mathbf{z})$ | Mixture density network that predicts the parameters of a Gaussian mixture. |
| $\mathbf{x} = \mathbf{h}_{\boldsymbol{\theta}_c}(\mathbf{z}, \mathbf{u})$ | Transfer or sampling function corresponding to the channel conditional density. |
| $\mathbf{f}_{\boldsymbol{\theta}}(\mathbf{1}_y) = \mathbf{D}_{\boldsymbol{\theta}_d}(\mathbf{h}_{\boldsymbol{\theta}_c}(\mathbf{E}_{\boldsymbol{\theta}_e}(\mathbf{1}_y), \mathbf{u}))$ | Input-output mapping of the autoencoder with combined parameter vector $\boldsymbol{\theta}^T = [\boldsymbol{\theta}_e^T, \boldsymbol{\theta}_c^T, \boldsymbol{\theta}_d^T]$. |
| $\boldsymbol{\psi}^T = [\boldsymbol{\psi}_1^T, \cdots, \boldsymbol{\psi}_k^T]$ | Affine transformation (adaptation) parameters per component used to adapt the MDN. |
| $\mathbf{g}_{\mathbf{z}i}$ and $\mathbf{g}_{\mathbf{z}i}^{-1},\ i \in [k], \mathbf{z} \in \mathcal{Z}$ | Affine transformations between the components of the source-to-target Gaussian mixtures and vice-verse. |
| $D_{\mathrm{KL}}(p, q)$ | Kullback-Leibler divergence between the distributions $p$ and $q$. |
| $N(\cdot \,\|\, \boldsymbol{\mu}, \boldsymbol{\Sigma})$ | Multivariate Gaussian density with mean vector $\boldsymbol{\mu}$ and covariance matrix $\boldsymbol{\Sigma}$. |
| $\delta(\mathbf{x} - \mathbf{x}_0)$ | Dirac delta or impulse function centered at $\mathbf{x}_0$. |
| $\mathrm{Cat}(p_1, \cdots, p_k)$ | Categorical distribution with $p_i \geq 0$ and $\sum_i p_i = 1$. |
| $\mathbb{1}(c)$ | Indicator function mapping a predicate $c$ to 1 if true and 0 if false. |
| $\|\mathbf{x}\|_p$ | $\ell_p$ norm of a vector $\mathbf{x}$. |

The appendices are organized as follows:

- Appendix A discusses the theoretical results from the main paper.
- Appendix B provides additional details on the proposed method including:
  - Discussion on class labels and labeled data in the communication setting (Appendix B.1).
  - Feature and parameter transformation between multivariate Gaussians (Appendix B.2).
  - Generative adaptation of the MDN channel (Appendix B.3).
  - The validation metric used for setting the hyper-parameter $\lambda$ (Appendix B.4).
  - Computational complexity analysis of the proposed method (Appendix B.5).
  - Limitations and future work (Appendix B.6).
- Appendix C provides additional details on the experiments and additional results, including ablation studies of the proposed method.
- Appendix D provides additional background on the following topics: 1) components of an end-to-end autoencoder-based communication system, 2) generative modeling using mixture density networks, 3) training algorithm of the autoencoder, and 4) a primer on domain adaptation.
- Appendix E provides details on the MDN training and differentiable sampling using the Gumbel-softmax reparametrization.
- Appendix F provides details on the simulated channel distributions used in our experiments.

## A  THEORETICAL RESULTS

**Propostion 1 (restatement).** The joint distributions $p(\mathbf{x}, y, \mathbf{z})$ and $p(\mathbf{x}, y)$ can be expressed in the following form:

$$p(\mathbf{x}, y, \mathbf{z}) \;=\; p\big(\mathbf{x} \,|\, \mathbf{E}_{\boldsymbol{\theta}_e}(\mathbf{1}_y)\big)\, p(y)\, \delta(\mathbf{z} - \mathbf{E}_{\boldsymbol{\theta}_e}(\mathbf{1}_y)), \;\; \forall \mathbf{x}, \mathbf{z} \in \mathbb{R}^d,\, y \in \mathcal{Y}$$

$$p(\mathbf{x}, y) \;=\; p\big(\mathbf{x} \,|\, \mathbf{E}_{\boldsymbol{\theta}_e}(\mathbf{1}_y)\big)\, p(y), \;\; \forall \mathbf{x} \in \mathbb{R}^d,\, y \in \mathcal{Y}, \tag{9}$$

where $\delta(\cdot)$ is the Dirac delta (or Impulse) function, and we define $p(\mathbf{x} \,|\, y) := p\big(\mathbf{x} \,|\, \mathbf{E}_{\boldsymbol{\theta}_e}(\mathbf{1}_y)\big)$ as the conditional distribution of $\mathbf{x}$ given the class $y$.

**Proof.** It follows from the dependence $y \to \mathbf{z} \to \mathbf{x}$ defined by our generative model that

$$p(\mathbf{x}, y, \mathbf{z}) \;=\; p(y)\, p(\mathbf{z} \,|\, y)\, p(\mathbf{x} \,|\, \mathbf{z}, y)$$
$$=\; p(y)\, \delta(\mathbf{z} - \mathbf{E}_{\boldsymbol{\theta}_e}(\mathbf{1}_y))\, p(\mathbf{x} \,|\, \mathbf{E}_{\boldsymbol{\theta}_e}(\mathbf{1}_y), y)$$
$$=\; p(y)\, \delta(\mathbf{z} - \mathbf{E}_{\boldsymbol{\theta}_e}(\mathbf{1}_y))\, p(\mathbf{x} \,|\, \mathbf{E}_{\boldsymbol{\theta}_e}(\mathbf{1}_y)).$$

In the second step, the conditional $p(\mathbf{z} \,|\, y)$ reduces to the Dirac delta since the symbol $\mathbf{z}$ can only take one of the $m$ values from the constellation $\mathcal{Z} = \{\mathbf{E}_{\boldsymbol{\theta}_e}(\mathbf{1}_1), \cdots, \mathbf{E}_{\boldsymbol{\theta}_e}(\mathbf{1}_m)\}$ (for a fixed encoder mapping). The distribution $p(\mathbf{x}, y)$ in Eq. (9) is obtained from the third step by integrating $p(\mathbf{x}, y, \mathbf{z})$ over all $\mathbf{z}$, and using the integration property of the Dirac delta.

### A.1 KL-DIVERGENCE BETWEEN THE SOURCE AND TARGET GAUSSIAN MIXTURES

**Propostion 2 (restatement).** Given $m$ Gaussian mixtures from the source domain and $m$ Gaussian mixtures from the target domain (one each per class), which satisfy Assumptions A1 and A2, the KL-divergence between $P_{\boldsymbol{\theta}_c}(\mathbf{x}, K \,|\, \mathbf{z})$ and $P_{\widehat{\boldsymbol{\theta}}_c}(\mathbf{x}, K \,|\, \mathbf{z})$ can be computed in closed-form, and is given by:

$$
\overline{D}_\psi(P_{\boldsymbol{\theta}_c}, P_{\widehat{\boldsymbol{\theta}}_c}) \;=\; \mathbb{E}_{P_{\boldsymbol{\theta}_c}}\left[\log \frac{P_{\boldsymbol{\theta}_c}(\mathbf{x}, K \,|\, \mathbf{z})}{P_{\widehat{\boldsymbol{\theta}}_c}(\mathbf{x}, K \,|\, \mathbf{z})}\right] \;=\; \sum_{\mathbf{z} \in \mathcal{Z}} p(\mathbf{z}) \sum_{i=1}^{k} \pi_i(\mathbf{z})\, \log \frac{\pi_i(\mathbf{z})}{\widehat{\pi}_i(\mathbf{z})}
$$

$$
+\; \sum_{\mathbf{z} \in \mathcal{Z}} p(\mathbf{z}) \sum_{i=1}^{k} \pi_i(\mathbf{z})\, D_{\mathrm{KL}}\Big(N\big(\,\cdot\,|\, \boldsymbol{\mu}_i(\mathbf{z}), \boldsymbol{\Sigma}_i(\mathbf{z})\big),\, N\big(\,\cdot\,|\, \widehat{\boldsymbol{\mu}}_i(\mathbf{z}), \widehat{\boldsymbol{\Sigma}}_i(\mathbf{z})\big)\Big), \tag{10}
$$

where $K$ is the mixture component random variable. The first term is the KL-divergence between the component prior probabilities, which simplifies into a function of the parameters $[\beta_1, \gamma_1, \cdots, \beta_k, \gamma_k]$. The second term involves the KL-divergence between two multivariate Gaussians (a standard result), which also simplifies into a function of $\psi$.

**Proof.** Referring to § 3.1, we derive the closed-form KL-divergence between the source and target Gaussian mixtures under Assumptions 1 and 2, *i.e.*, the source and target Gaussian mixtures have the same number of components that have a one-to-one association. Recall that $\boldsymbol{\theta}_c$ and $\widehat{\boldsymbol{\theta}}_c$ are the parameters of the original (source) and the adapted (target) MDN respectively. Let $K \in \{1, \cdots, k\}$ denote the latent component random variable.

$$
\overline{D}_\psi(P_{\boldsymbol{\theta}_c}, P_{\widehat{\boldsymbol{\theta}}_c}) \;=\; \mathbb{E}_{P_{\boldsymbol{\theta}_c}}\left[\log \frac{P_{\boldsymbol{\theta}_c}(\mathbf{x}, K \,|\, \mathbf{z})}{P_{\widehat{\boldsymbol{\theta}}_c}(\mathbf{x}, K \,|\, \mathbf{z})}\right]
$$

$$
=\; \sum_{\mathbf{z} \in \mathcal{Z}} p(\mathbf{z}) \sum_{i=1}^{k} \int_{\mathbb{R}^d} P_{\boldsymbol{\theta}_c}(\mathbf{x}, K = i \,|\, \mathbf{z})\, \log \frac{P_{\boldsymbol{\theta}_c}(\mathbf{x}, K = i \,|\, \mathbf{z})}{P_{\widehat{\boldsymbol{\theta}}_c}(\mathbf{x}, K = i \,|\, \mathbf{z})}\, d\mathbf{x}
$$

$$
=\; \sum_{\mathbf{z} \in \mathcal{Z}} p(\mathbf{z}) \sum_{i=1}^{k} P_{\boldsymbol{\theta}_c}(K = i \,|\, \mathbf{z}) \int_{\mathbb{R}^d} P_{\boldsymbol{\theta}_c}(\mathbf{x} \,|\, \mathbf{z}, K = i)\, \log \frac{P_{\boldsymbol{\theta}_c}(K = i \,|\, \mathbf{z})\, P_{\boldsymbol{\theta}_c}(\mathbf{x} \,|\, \mathbf{z}, K = i)}{P_{\widehat{\boldsymbol{\theta}}_c}(K = i \,|\, \mathbf{z})\, P_{\widehat{\boldsymbol{\theta}}_c}(\mathbf{x} \,|\, \mathbf{z}, K = i)}\, d\mathbf{x}
$$

$$
=\; \sum_{\mathbf{z} \in \mathcal{Z}} p(\mathbf{z}) \sum_{i=1}^{k} \pi_i(\mathbf{z}) \int_{\mathbb{R}^d} N\left(\mathbf{x} \,|\, \boldsymbol{\mu}_i(\mathbf{z}), \boldsymbol{\Sigma}_i(\mathbf{z})\right) \left(\log \frac{\pi_i(\mathbf{z})}{\widehat{\pi}_i(\mathbf{z})} + \log \frac{N\left(\mathbf{x} \,|\, \boldsymbol{\mu}_i(\mathbf{z}), \boldsymbol{\Sigma}_i(\mathbf{z})\right)}{N\left(\mathbf{x} \,|\, \widehat{\boldsymbol{\mu}}_i(\mathbf{z}), \widehat{\boldsymbol{\Sigma}}_i(\mathbf{z})\right)}\right) d\mathbf{x}
$$

$$
=\; \sum_{\mathbf{z} \in \mathcal{Z}} p(\mathbf{z}) \sum_{i=1}^{k} \pi_i(\mathbf{z})\, \log \frac{\pi_i(\mathbf{z})}{\widehat{\pi}_i(\mathbf{z})}
$$

$$
+\; \sum_{\mathbf{z} \in \mathcal{Z}} p(\mathbf{z}) \sum_{i=1}^{k} \pi_i(\mathbf{z})\, D_{\mathrm{KL}}\left(N\left(\,\cdot\,|\, \boldsymbol{\mu}_i(\mathbf{z}), \boldsymbol{\Sigma}_i(\mathbf{z})\right), N\left(\,\cdot\,|\, \widehat{\boldsymbol{\mu}}_i(\mathbf{z}), \widehat{\boldsymbol{\Sigma}}_i(\mathbf{z})\right)\right). \tag{11}
$$

The second term in the final expression involves the KL-divergence between two multivariate Gaussians (a standard result) given by

$$
D_{\mathrm{KL}}\left(N(\,\cdot\,|\, \boldsymbol{\mu}, \boldsymbol{\Sigma}), N(\,\cdot\,|\, \widehat{\boldsymbol{\mu}}, \widehat{\boldsymbol{\Sigma}})\right) \;=\; \frac{1}{2} \log \frac{\det(\widehat{\boldsymbol{\Sigma}})}{\det(\boldsymbol{\Sigma})} + \frac{1}{2} \operatorname{tr}(\widehat{\boldsymbol{\Sigma}}^{-1} \boldsymbol{\Sigma})
$$

$$
+\; \frac{1}{2}(\widehat{\boldsymbol{\mu}} - \boldsymbol{\mu})^T \widehat{\boldsymbol{\Sigma}}^{-1}(\widehat{\boldsymbol{\mu}} - \boldsymbol{\mu}) - \frac{d}{2}.
$$

For clarity, we further simplify Eq. (11) for the case of diagonal covariances by applying the above result. Recall that the Gaussian mixture parameters of the source and target domains are related by the parameter transformations in Eq. (4). The second term in Eq. (11) involving the KL-divergence between multivariate Gaussians, simplifies to

$$
D_{\mathrm{KL}}\left(N\left(\,\cdot\,|\, \boldsymbol{\mu}_i(\mathbf{z}), \boldsymbol{\sigma}_i^2(\mathbf{z})\right), N\left(\,\cdot\,|\, \widehat{\boldsymbol{\mu}}_i(\mathbf{z}), \widehat{\boldsymbol{\sigma}}_i^2(\mathbf{z})\right)\right)
$$

$$
=\; \frac{1}{2} \sum_{j=1}^{d} \left[\log c_{ij}^2 + \frac{1}{c_{ij}^2} + \frac{1}{c_{ij}^2\, \sigma_{ij}^2(\mathbf{z})}\left(a_{ij}\, \mu_{ij}(\mathbf{z}) + b_{ij} - \mu_{ij}(\mathbf{z})\right)^2\right] - \frac{d}{2}. \tag{12}
$$

The first term in Eq. (11) involving the KL-divergence between the component prior probabilties can be expressed as a function of the adaptation parameters $[\beta_1, \gamma_1, \cdots, \beta_k, \gamma_k]$ as follows:

$$
\sum_{i=1}^{k} \pi_i(\mathbf{z}) \log \frac{\pi_i(\mathbf{z})}{\widehat{\pi}_i(\mathbf{z})} = \sum_{i=1}^{k} \frac{e^{\alpha_i(\mathbf{z})}}{q(\mathbf{z})} \left[ \log \frac{e^{\alpha_i(\mathbf{z})}}{q(\mathbf{x})} - \log \frac{e^{\beta_i \, \alpha_i(\mathbf{z}) + \gamma_i}}{\widehat{q}(\mathbf{z})} \right]
$$

$$
= \log\left(\sum_{i=1}^{k} e^{\beta_i \, \alpha_i(\mathbf{z}) + \gamma_i}\right) - \log\left(\sum_{i=1}^{k} e^{\alpha_i(\mathbf{z})}\right) + \sum_{i=1}^{k} \frac{e^{\alpha_i(\mathbf{z})}}{q(\mathbf{z})} \left( \alpha_i(\mathbf{z}) - \beta_i \, \alpha_i(\mathbf{z}) - \gamma_i \right), \quad (13)
$$

where $q(\mathbf{z}) = \sum_{j=1}^{k} e^{\alpha_j(\mathbf{z})}$ and $\widehat{q}(\mathbf{x}) = \sum_{j=1}^{k} e^{\beta_j \, \alpha_j(\mathbf{z}) + \gamma_j}$ are the normalization terms in the softmax function. Substituting Eqs. (12) and (13) into the last step of Eq. (11) gives the KL-divergence between the source and target Gaussian mixtures as a function of the adaptation parameters $\boldsymbol{\psi}$.

## A.2 OPTIMALITY OF THE FEATURE TRANSFORMATION

We show that the proposed feature transformation at the decoder in § 3.3 is optimal in the mimimum mean-squared error sense. The problem setting is that, at the decoder, we observe an input $\mathbf{x}^t$ from the target domain marginal distribution, *i.e.*,

$$
\mathbf{x}^t \sim p^t(\mathbf{x}) = \sum_{\mathbf{z} \in \mathcal{Z}} p(\mathbf{z}) \sum_{i=1}^{k} \widehat{\pi}_i(\mathbf{z}) \, N\big(\mathbf{x} \,|\, \widehat{\boldsymbol{\mu}}_i(\mathbf{z}), \widehat{\boldsymbol{\Sigma}}_i(\mathbf{z})\big),
$$

where $\mathcal{Z} = \{\mathbf{E}_{\boldsymbol{\theta}_e}(\mathbf{1}_1), \cdots, \mathbf{E}_{\boldsymbol{\theta}_e}(\mathbf{1}_m)\}$ is the encoder's constellation. Suppose we knew the symbol $\mathbf{z} = \mathbf{E}_{\boldsymbol{\theta}_e}(\mathbf{1}_y)$ that was transmitted and the mixture component $i \in [k]$, then the transformation $\mathbf{g}_{\mathbf{z}i}^{-1}(\mathbf{x}^t)$ in Eq. (7) can map $\mathbf{x}^t$ to the corresponding Gaussian component of the source distribution. However, since $\mathbf{z}$ and $i$ are not observed at the decoder, we propose to find the transformation $\mathbf{g}^{-1} : \mathbb{R}^d \mapsto \mathbb{R}^d$ (independent of $\mathbf{z}$ and $i$) that minimizes the following expected squared error:

$$
J\big(\mathbf{g}^{-1}(\mathbf{x}^t)\big) = \frac{1}{2} \mathbb{E}_{P_{\widehat{\boldsymbol{\theta}}_c}(\mathbf{z}, i \,|\, \mathbf{x})} \big[ \|\mathbf{g}_{\mathbf{z}i}^{-1}(\mathbf{x}^t) - \mathbf{g}^{-1}(\mathbf{x}^t)\|_2^2 \,|\, \mathbf{x}^t \big]. \quad (14)
$$

This is the conditional expectation over $(\mathbf{z}, i)$ given $\mathbf{x}^t$ with respect to the posterior distribution $P_{\widehat{\boldsymbol{\theta}}_c}(\mathbf{z}, i \,|\, \mathbf{x})$. Since $\mathbf{x}^t$ is fixed, the above objective is a function of the vector $\mathbf{w} := \mathbf{g}^{-1}(\mathbf{x}^t) \in \mathbb{R}^d$, and it can be simplified as follows:

$$
\begin{aligned}
J(\mathbf{w}) &= \frac{1}{2} \mathbb{E}_{P_{\widehat{\boldsymbol{\theta}}_c}(\mathbf{z}, i \,|\, \mathbf{x})} \big[ \|\mathbf{g}_{\mathbf{z}i}^{-1}(\mathbf{x}^t) - \mathbf{w}\|_2^2 \,|\, \mathbf{x}^t \big] \\
&= \frac{1}{2} \mathbb{E}_{P_{\widehat{\boldsymbol{\theta}}_c}(\mathbf{z}, i \,|\, \mathbf{x})} \big[ \mathbf{g}_{\mathbf{z}i}^{-1}(\mathbf{x}^t)^T \mathbf{g}_{\mathbf{z}i}^{-1}(\mathbf{x}^t) \,|\, \mathbf{x}^t \big] + \frac{1}{2} \mathbf{w}^T \mathbf{w} \\
&\quad - \mathbf{w}^T \mathbb{E}_{P_{\widehat{\boldsymbol{\theta}}_c}(\mathbf{z}, i \,|\, \mathbf{x})} \big[ \mathbf{g}_{\mathbf{z}i}^{-1}(\mathbf{x}^t) \,|\, \mathbf{x}^t \big].
\end{aligned}
$$

Note that $\mathbf{w}$ comes outside the expectation since it does not depend on $\mathbf{z}$ or $i$. The minimum of this simple quadratic function can be found by setting the gradient of $J$ with respect to $\mathbf{w}$ to $\mathbf{0}$, giving

$$
\begin{aligned}
\mathbf{w}^\star = \mathbf{g}^{-1}(\mathbf{x}^t) &= \mathbb{E}_{P_{\widehat{\boldsymbol{\theta}}_c}(\mathbf{z}, i \,|\, \mathbf{x})} \big[ \mathbf{g}_{\mathbf{z}i}^{-1}(\mathbf{x}^t) \,|\, \mathbf{x}^t \big] \\
&= \sum_{\mathbf{z} \in \mathcal{Z}} \sum_{i \in [k]} P_{\widehat{\boldsymbol{\theta}}_c}(\mathbf{z}, i \,|\, \mathbf{x}^t) \, \mathbf{g}_{\mathbf{z}i}^{-1}(\mathbf{x}^t).
\end{aligned}
$$

This is the feature transformation at the decoder proposed in § 3.3.

## B ADDITIONAL DETAILS ON THE PROPOSED METHOD

In this section we provide additional details on the proposed method that could not be discussed in § 3 of the main paper.

### B.1 CLASS LABELS AND LABELED DATA

We would like to clarify that the statement "class labels are available for free" is made in Section 3 in order to highlight the fact that class labels are easy to obtain in this end-to-end communication

setting, unlike other domains (e.g. computer vision) where labeling data could be expensive. Since the transmitted message is also the class label, it is always available without additional effort during the data collection (from the packet preambles). However, note that it is still challenging / expensive to collect a large number of samples for domain adaptation, as discussed in Section 1. In contrast, it may be easy to obtain plenty of unlabeled data in other domains such as computer vision, where labeling is expensive.

In communication protocols, preambles are attached to the front of the packets for synchronization, carrier frequency offset correction, and other tasks. The preambles consist of sequences of known symbols (which have a one-to-one mapping to the messages). Therefore, these sequences can be used as the labeled dataset since the receiver obtains the distorted symbol and knows the ground truth. The proposed MDN adaptation and input transformation at the decoder do not incur any modifications to the encoder (transmitter side). The constellation learned by the autoencoder is kept fixed during adaptation. Therefore, using the preambles from a small number of packets, our method performs adaptation at the receiver side and maintains the symbol error rate performance without communicating any information back to the encoder.

### B.2 TRANSFORMATION BETWEEN MULTIVARIATE GAUSSIANS

We discuss the feature and parameter transformations between any two multivariate Gaussians. This result was applied to formulate the MDN adaptation in Eqs. (4) and (7). Consider first the standard transformation from $\mathbf{x} \sim N(\cdot \,|\, \boldsymbol{\mu}, \boldsymbol{\Sigma})$ to $\widehat{\mathbf{x}} \sim N(\cdot \,|\, \widehat{\boldsymbol{\mu}}, \widehat{\boldsymbol{\Sigma}})$ given by the two-step process:

- Apply a whitening transformation $\mathbf{z} = \mathbf{D}^{-1/2}\,\mathbf{U}^T\,(\mathbf{x} - \boldsymbol{\mu})$ such that $\mathbf{z} \sim N(\cdot \,|\, \mathbf{0}, \mathbf{I})$.
- Transform $\mathbf{z}$ into the new Gaussian density using $\widehat{\mathbf{x}} = \widehat{\mathbf{U}}\,\widehat{\mathbf{D}}^{1/2}\,\mathbf{z} + \widehat{\boldsymbol{\mu}}$.

We have denoted the eigen-decomposition of the covariance matrices by $\boldsymbol{\Sigma} = \mathbf{U}\mathbf{D}\mathbf{U}^T$ and $\widehat{\boldsymbol{\Sigma}} = \widehat{\mathbf{U}}\widehat{\mathbf{D}}\widehat{\mathbf{U}}^T$, where $\mathbf{U}$ and $\widehat{\mathbf{U}}$ are the orthonormal eigenvector matrices, and $\mathbf{D}$ and $\widehat{\mathbf{D}}$ are the diagonal eigenvalue matrices. Combining the two steps, the overall transformation from $\mathbf{x}$ to $\widehat{\mathbf{x}}$ is given by

$$\widehat{\mathbf{x}} = \widehat{\mathbf{U}}\,\widehat{\mathbf{D}}^{1/2}\,\mathbf{D}^{-1/2}\,\mathbf{U}^T\,(\mathbf{x} - \boldsymbol{\mu}) + \widehat{\boldsymbol{\mu}}. \tag{15}$$

Suppose we define the matrix $\mathbf{C} = \widehat{\mathbf{U}}\,\widehat{\mathbf{D}}^{1/2}\,\mathbf{D}^{-1/2}\,\mathbf{U}^T$, then it is easily verified that the covariance matrices are related by $\widehat{\boldsymbol{\Sigma}} = \mathbf{C}\,\boldsymbol{\Sigma}\,\mathbf{C}^T$. In general, the mean vector and covariance matrix of any two Gaussians can be related by the following parameter transformations:

$$\widehat{\boldsymbol{\mu}} = \mathbf{A}\,\boldsymbol{\mu} + \mathbf{b} \quad \text{and} \quad \widehat{\boldsymbol{\Sigma}} = \mathbf{C}\,\boldsymbol{\Sigma}\,\mathbf{C}^T, \tag{16}$$

with parameters $\mathbf{A} \in \mathbb{R}^{d \times d}$, $\mathbf{b} \in \mathbb{R}^d$, and $\mathbf{C} \in \mathbb{R}^{d \times d}$. Substituting the above parameter transformations into the feature transformation in Eq. (15), we get

$$\widehat{\mathbf{x}} = \mathbf{C}\,(\mathbf{x} - \boldsymbol{\mu}) + \mathbf{A}\,\boldsymbol{\mu} + \mathbf{b}.$$

From the above, we can also define the inverse feature transformation from $\widehat{\mathbf{x}} \sim N(\cdot \,|\, \widehat{\boldsymbol{\mu}}, \widehat{\boldsymbol{\Sigma}})$ to $\mathbf{x} \sim N(\cdot \,|\, \boldsymbol{\mu}, \boldsymbol{\Sigma})$:

$$\mathbf{x} = \mathbf{C}^{-1}\,(\widehat{\mathbf{x}} - \mathbf{A}\,\boldsymbol{\mu} - \mathbf{b}) + \boldsymbol{\mu}.$$

### B.3 GENERATIVE ADAPTATION OF THE MDN

In § 3.2, we discussed the discriminative adaptation objective for the MDN, which is used when the MDN is adapted as part of the autoencoder in order to improve the end-to-end error rate. This adaptation approach was used for the experiments in § 4. On the other hand, we may be interested in adapting the MDN in isolation with the goal of improving its performance as a generative model of the channel. For this scenario, the adaptation objective Eq. 6 is modified as follows. The first (data-dependent) term is replaced with the negative conditional log-likelihood (CLL) of the target dataset, while the second KL-divergence term remains the same:

$$J_{\text{CLL}}(\boldsymbol{\psi}\,;\lambda) = \frac{-1}{N^t} \sum_{n=1}^{N^t} \log P_{\widehat{\boldsymbol{\theta}}_c}(\mathbf{x}_n^t \,|\, \mathbf{z}_n^t) + \lambda\,\overline{D}_{\boldsymbol{\psi}}(P_{\boldsymbol{\theta}_c}, P_{\widehat{\boldsymbol{\theta}}_c}), \tag{17}$$

where $\widehat{\boldsymbol{\mu}}_i(\mathbf{z}), \widehat{\boldsymbol{\Sigma}}_i(\mathbf{z})$ and $\widehat{\alpha}_i(\mathbf{z})$ as a function of $\boldsymbol{\psi}$ are given by Eq. (4). The parameters of the original Gaussian mixture $\alpha_i(\mathbf{z}), \boldsymbol{\mu}_i(\mathbf{z}), \boldsymbol{\Sigma}_i(\mathbf{z}), \forall i$ are constants since they have no dependence on

$\psi$. The regularization constant $\lambda \geq 0$ controls the allowed KL-divergence between the source and target Gaussian mixtures. Small values of $\lambda$ weight the CLL term more, allowing more exploration in the adaptation, while large values of $\lambda$ impose a strong regularization to constrain the space of target distributions. We evaluate the performance of this generative MDN adaptation in Appendix C.6.

### B.4   Validation Metric For Automatically Setting $\lambda$

The choice of $\lambda$ in the adaptation objectives Eqs. (6) and 17 is crucial as it sets the right level of regularization suitable for the target domain distribution. Since the target domain dataset is very small, it is difficult to apply cross-validation type of methods to select $\lambda$. We propose a validation metric $V(\psi; \mathcal{D}^t)$ that utilizes the feature-transformed target domain dataset to evaluate the quality of the adapted solutions for different $\lambda$ values.

Let $\psi$ denote the adaptation parameters found by minimizing the objective Eq. (6) for a specific $\lambda \geq 0$. The feature transformation (from target to source domain) at the decoder $\mathbf{g}^{-1}(\mathbf{x})$ based on the adaptation parameters $\psi$ is given by Eq. (8). Recall that the target domain dataset is $\mathcal{D}^t = \{(\mathbf{x}_n^t, y_n^t, \mathbf{z}_n^t), \, n = 1, \cdots, N^t\}$. We define the feature-transformed target domain dataset as:

$$\mathcal{D}_{\text{trans}}^t = \{(\mathbf{g}^{-1}(\mathbf{x}_n^t), y_n^t, \mathbf{z}_n^t), \, n = 1, \cdots, N^t\}.$$

Suppose $\psi$ is a good adaptation solution, then we expect the decoder (trained on the source domain dataset) to have good classification performance on $\mathcal{D}_{\text{trans}}^t$. For a given feature-transformed target domain sample, the decoder predicts the class posterior probabilities: $\mathbf{D}_{\boldsymbol{\theta}_d}(\mathbf{g}^{-1}(\mathbf{x}_n^t)) = [P_{\boldsymbol{\theta}_d}(1 \mid \mathbf{g}^{-1}(\mathbf{x}_n^t)), \cdots, P_{\boldsymbol{\theta}_d}(m \mid \mathbf{g}^{-1}(\mathbf{x}_n^t))]$. We define the validation metric as the *negative posterior log-likelihood* of the decoder on $\mathcal{D}_{\text{trans}}^t$, given by

$$V(\psi; \mathcal{D}^t) = -\frac{1}{N^t} \sum_{n=1}^{N^t} \log P_{\boldsymbol{\theta}_d}\left(y_n^t \mid \mathbf{g}^{-1}(\mathbf{x}_n^t)\right) \tag{18}$$

We expect smaller values of $V(\psi; \mathcal{D}^t)$ to correspond to better adaptation solutions. The adaptation objective is minimized with $\lambda$ varied over a range of values, and in each case the adapted solution $\psi$ is evaluated using the validation metric. The pair of $\lambda$ and $\psi$ resulting in the smallest validation metric is chosen as the final adapted solution. The search set of $\lambda$ used in our experiments was $\{10^{-5}, 10^{-4}, 10^{-3}, 10^{-2}, 0.1, 1, 10, 100\}$. See Appendix C.4 for an ablation study on the choice of hyper-parameter $\lambda$ using this validation metric.

**Generative MDN Adaptation.** The validation metric proposed above depends on the decoder, and cannot be used when the MDN is adapted as a generative model in isolation (Appendix B.3). For this setting, we modify the validation metric based on the following idea. Suppose the adaptation finds a good solution, then we expect $\mathcal{D}_{\text{trans}}^t$ to have a high conditional log-likelihood under the (original) source domain MDN. The validation metric is therefore given by

$$V(\psi; \mathcal{D}^t) = -\frac{1}{N^t} \sum_{n=1}^{N^t} \log P_{\boldsymbol{\theta}_c}\left(\mathbf{g}^{-1}(\mathbf{x}_n^t) \mid \mathbf{z}_n^t\right), \tag{19}$$

where $P_{\boldsymbol{\theta}_c}$ is the Gaussian mixture given by Eq. 1.

### B.5   Complexity Analysis

We provide an analysis of the computational complexity of the proposed adaptation methods.

**MDN Adaptation.**

The number of free parameters being optimized in the adaptation objective (Eqs. 6 or 17) is given by $|\psi| = k(2d^2 + d + 2)$. This is much smaller than the number of parameters in a typical MDN, even considering only the final fully-connected layer (see Table 1 for a comparison). Each step of the BFGS optimization involves computing the objective function, its gradient, and an estimate of its inverse Hessian. The cost of one step of BFGS can thus be expressed as $O(N^t k d^2 |\psi|^2)$. Suppose BFGS runs for a maximum of $T$ iterations and the optimization is repeated for $L$ values of $\lambda$, then the overall cost of adaptation is given by $O(L T N^t k d^2 |\psi|^2)$. Note that the optimization for different $\lambda$ values can be easily solved in parallel.

**Test-time Adaptation at the Decoder.**

We analyze the computational cost of the feature transformation-based adaptation at the decoder proposed in § 3.3. Consider a single test input $\mathbf{x}^t$ at the decoder. The feature transformation method first computes the posterior distribution $P_{\widehat{\boldsymbol{\theta}}_c}(\mathbf{z}, i \mid \mathbf{x}^t)$ over the set of symbols-component pairs of size $k\,m$. Computation of each exponent factor in the posterior distribution requires $O(d^3)$ operations for the full-covariance case, and $O(d)$ operations for the diagonal covariance case. This corresponds to calculation of the log of the Gaussian density. Therefore, computation of the posterior distribution for a single $(\mathbf{z}, i)$ pair requires $O(k\,m\,d^3)$ operations for the full-covariance case (similarly for the diagonal case). Computation of the affine transformation $\mathbf{g}_{\mathbf{z}i}^{-1}(\mathbf{x}^t)$ for a single $(\mathbf{z}, i)$ pair requires $O(d^2)$ operations (the matrix $\mathbf{C}_i$ only needs to be inverted once prior to test-time adaptation). Since calculation of the posterior term dominates the computation, the overall cost of computing the transformation in Eq (8) over the $k\,m$ symbol-component pairs will be $O(k\,m\,k\,m\,d^3) = O(k^2\,m^2\,d^3)$.

We note that in practical communication systems $d$ is small (typically $d = 2$). The number of symbols or messages $m$ can vary from 4 to 1024 in powers of 2. The number of mixture components $k$ can be any positive integer, but is usually not more than a few tens to keep the size of the MDN practical. Therefore, the computational cost of test-time adaptation at the decoder based on the feature transformation method is relatively small, making our proposed adaptation very computationally efficient to implement at the receiver side of a communication system.

### B.6 LIMITATIONS AND FUTURE WORK

The proposed work focuses mainly on a mixture density network (MDN) as the generative channel model, which allows us to exploit some of their useful properties in our formulation. Generalizing the proposed few-shot domain adaptation to other types of generative channel models such as conditional GANs, VAEs, and normalizing flows (Dinh et al., 2017) could be an interesting direction. These generative models can handle more high-dimensional structured inputs.

The proposed work does not adapt the encoder network, *i.e.*, the autoencoder constellation is not adapted to changes in the channel distribution. Adapting the encoder, decoder, and channel networks jointly would allow for more flexibility, but would likely be slower and require more data from the target distribution.

We focused on memoryless channels, where inter-symbol-interference (ISI) is not a problem. In practice, communication channels can have memory and ISI would have to be addressed by the training and adaptation methods. Under changing channels, one would have to also adapt an Equalizer model (algorithm) in order to mitigate ISI.

## C ADDITIONAL EXPERIMENTS

We provide additional details on the experiments in § 4 and report additional results, including ablation studies on the proposed method.

### C.1 EXPERIMENTAL SETUP

We implemented the mixture density network and communication autoencoder models using TensorFlow (Abadi et al., 2015) and TensorFlow Probability. We used the BFGS optimizer implementation available in TensorFlow Probability. The code base for our work has been submitted as a supplementary material. All the experiments were run on a Macbook Pro with 16 GB memory and 8 CPU cores. Table 3 summarizes the architecture of the encoder, MDN (channel model), and decoder neural networks. Note that the output layer of the MDN is a concatenation (denoted by $\oplus$) of three fully-connected layers predicting the means, variances, and mixing prior logit parameters of the Gaussian mixture. The following setting is used in all our experiments. The size of the message set $m$ (also the number of classes) was fixed to 16, corresponding to 4 bits. The dimension of the encoding $d$ was set to 2, and the number of mixture components $k$ was set to 5. The size of the hidden layers $n_h$ was set to 100.

Table 3: Architecture of the Encoder, MDN channel, and Decoder neural networks. FC - fully connected (dense) layer; $\oplus$ denotes layer concatenation; ELU - exponential linear unit; $m$ - number of messages; $d$ - encoding dimension; $k$ - number of mixture components; $n_h$ - size of a hidden layer.

| Network | Layer | Activation |
|---|---|---|
| Encoder | FC, $m \times n_h$ | ReLU |
| | FC, $n_h \times d$ | Linear |
| | Normalization (avg. power) | None |
| MDN | FC, $d \times n_h$ | ReLU |
| | FC, $n_h \times n_h$ | ReLU |
| | FC, $n_h \times kd$ (means) | Linear |
| | $\oplus$ FC, $n_h \times kd$ (variances) | ELU $+ 1 + \epsilon$ |
| | $\oplus$ FC, $n_h \times k$ (prior logits) | Linear |
| Decoder | FC, $d \times n_h$ | ReLU |
| | FC, $n_h \times m$ | Softmax |

The parameters $\psi$ of the proposed adaptation method are initialized as follows for each component $i$:

$$\mathbf{A}_i = \mathbf{I}_d, \ \mathbf{b}_i = \mathbf{0}, \ \mathbf{C}_i = \mathbf{I}_d, \ \beta_i = 1, \ \gamma_i = 0,$$

where $\mathbf{I}_d$ is the $d \times d$ identity matrix. This initialization ensures that the target Gaussian mixtures (per class) are always initially equal to the source Gaussian mixtures. The regularization constant $\lambda$ in the adaptation objective was varied over 8 equally-spaced values on the $\log$-scale with range $10^{-5}$ to 100, specifically $\{10^{-5}, 10^{-4}, 10^{-3}, 10^{-2}, 0.1, 1, 10, 100\}$. The $\lambda$ value and $\psi$ corresponding to the smallest validation metric are selected as the final solution.

We used the Adam optimizer (Kingma & Ba, 2015) with a fixed learning rate of 0.001, batch size of 128, and 100 epochs for training the MDN. For adaptation of the MDN using the baseline methods *Finetune* and *Finetune last*, we used Adam with the same learning rate for 200 epochs. The batch size is set as $b = \max\{10, 0.1 N^t\}$, where $N^t$ is number of adaptation samples in the target dataset. For training the autoencoder using Algorithm 1, we found that stochastic gradient descent (SGD) with Nesterov momentum (constant 0.9), and an exponential learning rate schedule between 0.1 and 0.005 works better than Adam.

**Finetuning Baselines.** We provide additional details on the baselines *Finetune* and *Finetune last*. Both the methods first initialize the target domain MDN, encoder, and decoder networks with the corresponding parameters from the source domain. The method *Finetune* first finetunes all the MDN parameters to minimize the conditional log-likelihood of the target dataset using the Adam optimizer. After the MDN is finetuned, we freeze the parameters of the MDN and encoder, and train only the decoder using data generated from the updated MDN channel. The method *Finetune last* differs from *Finetune* in that it optimizes only the weights of the final MDN layer.

From the results in Figures 4, 5, and 6, we observe that the baselines *Finetune* and *Finetune last* have very similar performance compared to the case of no adaptation. We have investigated this carefully and verified that this is not due to a bug or insufficient optimization (*e.g.*, by checking if the final weights of the MDN and decoder are different for both methods). For both methods, we tried a range of learning rates for Adam and increased the number of epochs to a large number (beyond 200 was not helpful). We have reported the best-case results for these methods, which suggests that they are not effective at adaptation using small target domain datasets. As mentioned in Section 4.1, we hypothesize that using the KL-divergence based regularization and constraining the number of adaptation parameters leads to more effective performance of our method.

**Uncertainty Estimation.** Since there is inherent randomness in our experiments, especially with the small sample sizes of the target dataset, we always report average results from multiple trials. For the experiments on standard simulated channel variations (*e.g.*, AWGN to Ricean fading), we report the results from 10 trials. For the random Gaussian mixtures experiment, we report the average and standard error over 50 random source/target dataset pairs. For the FPGA experiments, we report the results from 20 random trials. The average metrics (symbol error rate and log-likelihood) are reported in the plots.

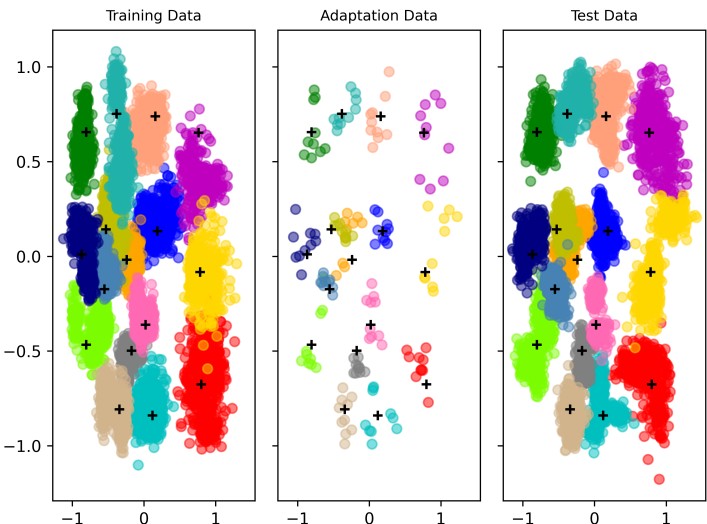

Figure 7: Example data plot for the case where the source and target domain class-conditional distributions are random Gaussian mixtures. The training data is from the source distribution, while the adaptation data (with 10 samples per class) and test data are from the target distribution.

**Evaluation Protocol.** We create a random class-stratified 50-50 train-test split (each of size 300,000) for data from both the source and target domains. Performance on both domains is always evaluated on the held-out test split. The train split from the target domain dataset is sub-sampled to create adaptation datasets of different sizes, specifically with 5, 10, 20, 30, 40, and 50 samples per class (symbol). For the generative adaptation experiments on the MDN (Appendix C.6), the number of adaptation samples from the target domain is reduced even further. We varied it from 2 samples per-class to 20 samples per-class in order to highlight the improvements obtained by the proposed method. The oracle baseline method, which retrains the autoencoder and MDN on the target distribution, uses the entire training dataset from the target domain.

**Choice of SNR.** For the experiments on simulated channel distributions such as AWGN, Ricean fading, and Uniform fading, we set the signal-to-noise ratio (SNR) to 14 dB for the source distribution and 20 dB for the target distribution. The connection between the SNR and the distribution parameters is given in Appendix F. We have experimented with other combinations of SNR for the source and target channels and found a similar trend in the adaptation performance.

In the simulated experiments, we focused on the SNR range of 14 dB to 20 dB. Our process for selecting this SNR range was by first evaluating the symbol error rate (SER) vs. SNR curve of the autoencoder for the different simulated channel distributions. We found that going below 14 dB SNR results in a degradation of the autoencoder's performance (except for the AWGN channel, which we do not use as a target distribution). Also, going above 20 dB SNR did not lead to a significant decrease in the SER. For the channels such as Ricean fading and Uniform fading, we found that even a retrained autoencoder has a relatively high error rate for lower SNRs.

### C.2 DETAILS ON THE FPGA EXPERIMENT

Referring to the experiment in § 4.2, for the real and over-the-air traces we used the platform from Lacruz et al. (2021). This ultra-wide-band mm-wave transceiver baseband memory-based design is developed on top of an ZCU111 RFSoC FPGA. This evaluation board features a Zynq Ultrascale + ZCU28DR. This FPGA is equipped with $8 \times 8$ AD/DA converters with Giga-sampling capabilities, which make it ideal for RF system development; the 4 GB DDR4 memories contain RF-ADCs with up to 4 GSPS of sampling rate, and RF-DACs with up to 6.544 GSPS. This board also includes a quad-core ARM Cortex-A53 and a dual-core ARM Cortex-R5 real-time processor.

For the radio frequency, we used 60 GHz RF front-end antennas. These kits include a $16 + 16$ TRX patch array antenna plus the RF module with up/down conversion from baseband to I/Q channels,

and TX/RX local oscillator (LO) frequency control. The antennas use $57 - 71$ GHz, a range of frequencies that cover the unlicensed 60 GHz band for mm-wave channels, and are managed from a PC Host via USB.

We implemented a hardware on the loop training. For the experimentation on real traces, we use Matlab as a central axis. The PC host running Matlab is connected to the platform via Ethernet. The FPGA can transmit different custom waveforms like 16-QAM frames from the 802.11ad and 802.11ay standards, with 2 GHz of bandwidth. The frames are sent over-the-air via 60 GHz radio frequency kits, and the samples are stored at the FPGA DDR memory. We decode the received data from the transmission, removing the preamble and header fields and extracting the symbols to train the MDN. We add a preamble to the generated constellation from the MDN for packet detection purposes, and we transmit again the new waveforms over-the-air. Finally, the adaptation is performed offline with the decoded symbols from the custom autoencoder-learned constellation.

**Source and Target Domains.**

For the experiment in § 4.2, we introduced distribution changes via IQ imbalance-based distortions to the symbol constellation, and evaluated the adaptation performance as a function of the level of imbalance. The source domain would be the original channel, the over-the-air link between the transmitter and receiver on which the training data is collected. This source domain data is used for training the MDN and the autoencoder. The target domain would be a modification of the source domain where the symbols used by the transmitter are distorted by modifying the in-phase and quadrature-phase (IQ) components of the RF signal. This causes a change in the distribution observed by the receiver (decoder), leading to a drop in performance without any adaptation.

### C.3 DETAILS ON THE RANDOM GAUSSIAN MIXTURE DATASETS

We created a simulated distribution shift setting where data from both the source and target domains are generated from class-conditional Gaussian mixtures whose parameters are modified between the two domains (*e.g.*, see Fig. 7). The parameters for the source and target Gaussian mixtures are generated as follows:

**Source Domain.** The source domain data is generated with a standard 16-QAM constellation $\mathcal{Z}_{\text{QAM}}$, which has 16 classes (messages). Let $k_s$ be the number of components in the source Gaussian mixture.

For each $\mathbf{z} \in \mathcal{Z}_{\text{QAM}}$:

- Calculate $d_{\min}$, the minimum distance from $\mathbf{z}$ to the remaining symbols in $\mathcal{Z}_{\text{QAM}}$. Let $\sigma_s = d_{\min} / 4$ be a constant standard deviation for this symbol.
- Component priors: generate $\pi_i(\mathbf{z}) \sim \text{Unif}(0.05, 0.95)$, $\forall i \in [k_s]$. Normalize the priors to sum to 1.
- Component means: generate $\boldsymbol{\mu}_i(\mathbf{z}) \sim N(\cdot \,|\, \mathbf{z}, \sigma_s^2 \mathbf{I})$, $\forall i \in [k_s]$.
- Component covariances: generate $s_1, \cdots, s_d \overset{\text{iid}}{\sim} \text{Unif}(0.2\,\sigma_s, \sigma_s)$ and let $\boldsymbol{\Sigma}_i(\mathbf{z}) = \text{diag}(s_1^2, \cdots, s_d^2)$, $\forall i \in [k_s]$ (the covariances are diagonal).
- Generate $N^s / m$ samples corresponding to symbol $\mathbf{z}$ from the Gaussian mixture: $\mathbf{x}_n^s \sim \sum_{i=1}^{k_s} \pi_i(\mathbf{z}) N(\mathbf{x} \,|\, \boldsymbol{\mu}_i(\mathbf{z}), \boldsymbol{\Sigma}_i(\mathbf{z}))$.

**Target Domain.** The parameters of the target Gaussian mixture are generated in a very similar way. The MDN and autoencoder are trained on the source domain dataset. Let $\mathcal{Z} = \{\mathbf{E}_{\boldsymbol{\theta}_e}(\mathbf{1}_1), \cdots, \mathbf{E}_{\boldsymbol{\theta}_e}(\mathbf{1}_m)\}$ be the constellation learned by the autoencoder. Let $k_t$ be the number of components in the target Gaussian mixture.

For each $\mathbf{z} \in \mathcal{Z}$:

- Calculate $d_{\min}$, the minimum distance from $\mathbf{z}$ to the remaining symbols in $\mathcal{Z}$. Let $\sigma_t = d_{\min} / 4$ be a constant standard deviation for this symbol.
- Component priors: generate $\widehat{\pi}_i(\mathbf{z}) \sim \text{Unif}(0.05, 0.95)$, $\forall i \in [k_t]$. Normalize the priors to sum to 1.
- Component means: generate $\widehat{\boldsymbol{\mu}}_i(\mathbf{z}) \sim N(\cdot \,|\, \mathbf{z}, \sigma_t^2 \mathbf{I})$, $\forall i \in [k_t]$.

- Component covariances: generate $s_1, \cdots, s_d \overset{\text{iid}}{\sim} \text{Unif}(0.2\,\sigma_t, \sigma_t)$ and let $\widehat{\boldsymbol{\Sigma}}_i(\mathbf{z}) = \text{diag}(s_1^2, \cdots, s_d^2),\ \forall i \in [k_t]$ (the covariances are diagonal).

- Generate $N^t / m$ samples corresponding to symbol $\mathbf{z}$ from the Gaussian mixture: $\mathbf{x}_n^t \sim \sum_{i=1}^{k_t} \widehat{\pi}_i(\mathbf{z})\, N(\mathbf{x}\,|\,\widehat{\boldsymbol{\mu}}_i(\mathbf{z}), \widehat{\boldsymbol{\Sigma}}_i(\mathbf{z}))$.

We set $k_s = k_t = 3$, except for the experiment where the source and target Gaussian mixtures are mismatched. In this case, $k_s$ and $k_t$ are randomly selected for each dataset from the range $\{3, 4, 5, 6\}$.

**Random Phase Shift.** We allow the channel output $\mathbf{x}$ to be randomly phase shifted on top of other distribution changes. This is done by matrix multiplication of $\mathbf{x}$ with a rotation matrix, where the rotation angle for each sample is uniformly selected from $[-\phi, \phi]$. We set $\phi$ to $\pi/18$ or 10 degrees. Results on a dataset with random phase shift applied on top of random Gaussian mixture distribution shift can be found in Fig. 5c.

## C.4 ABLATION EXPERIMENTS

We perform ablation experiments to understand: **1)** the choice of the hyper-parameter $\lambda$, **2)** the importance of the KL-divergence regularization in the adaptation objective, **3)** performance of our method when the source and target Gaussian mixtures have mismatched components, and **4)** the performance of our method when there is no distribution change.

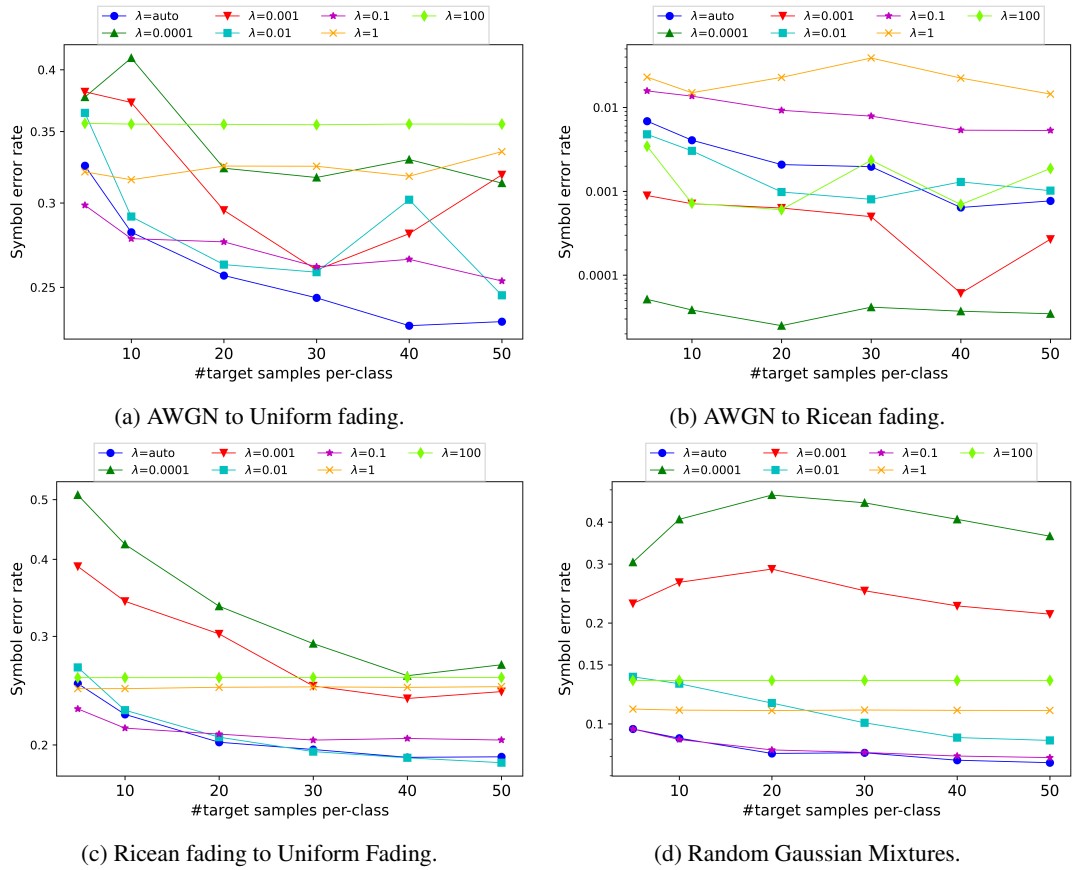

Figure 8: Evaluation of the validation metric for selection of the hyper-parameter $\lambda$ under different distribution shifts. The setting $\lambda = \text{auto}$ corresponds to automatic choice using the validation metric.

**Automatic Selection of Hyper-parameter $\lambda$.** We evaluate the proposed validation metric for automatically selecting the hyper-parameter $\lambda$ and report the results in Fig. 8. We run the proposed method for different fixed values of $\lambda$ as well as the automatically-selected $\lambda$, and compare their

performance on the target domain test set. We consider both simulated channel variations and the random Gaussian mixture datasets. From the figure, we observe that in most cases performance based on the automatically set value of $\lambda$ is better than other fixed choices of $\lambda$. The case of adaptation from AWGN to Ricean fading is an exception, where our method does not learn a good adaptation solution (see Fig. 4c). In this case, we observe from Fig. 8b that the setting $\lambda = 0.0001$ has the best symbol error rate.

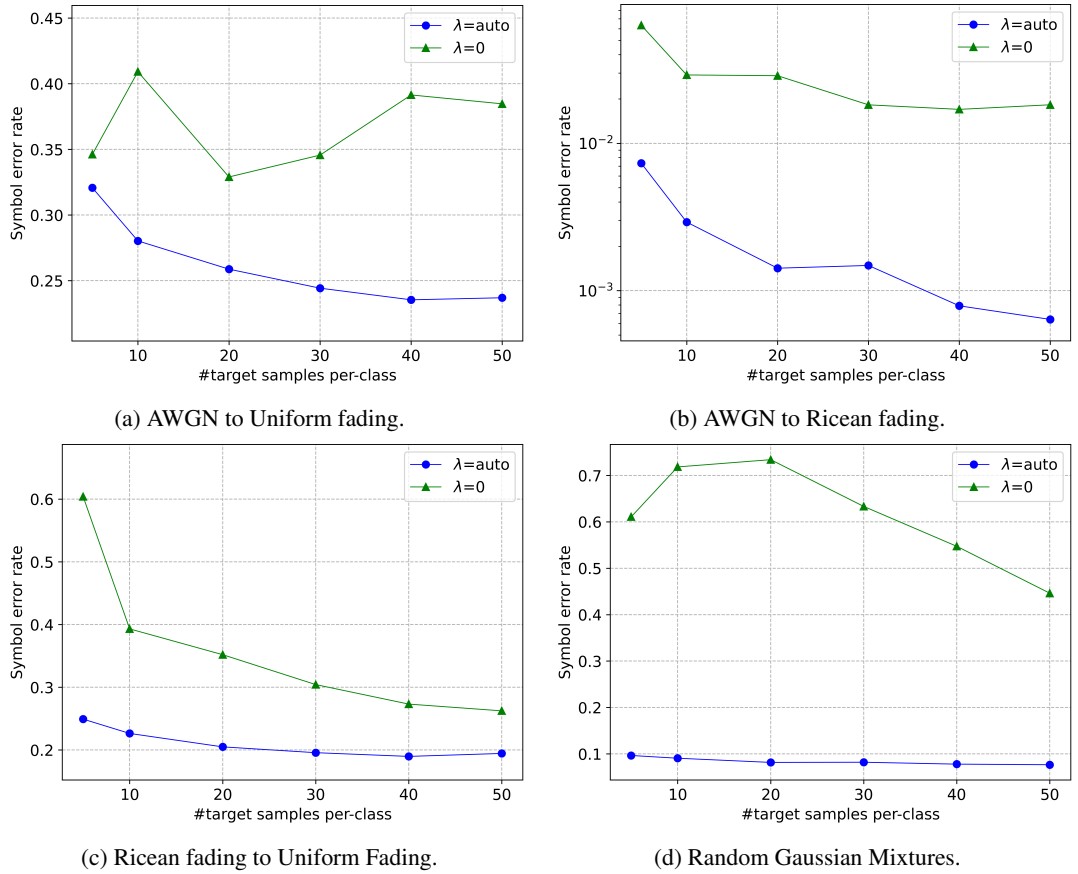

(a) AWGN to Uniform fading.

(b) AWGN to Ricean fading.

(c) Ricean fading to Uniform Fading.

(d) Random Gaussian Mixtures.

Figure 9: Evaluating the importance of the KL-divergence term in the adaptation objective. The setting $\lambda = $ auto corresponds to automatic choice using the validation metric.

**Performance Under Component Mismatch.** We evaluate the symbol error rate performance of all the methods in the setting where the number of components in the source and target Gaussian mixtures is mismatched. The number of components in the source and target Gaussian mixtures is randomly selected from the range 3 to 6. From Fig. 11, we observe that the proposed method has strong performance improvements even in this mismatched setting, suggesting that our method can perform well even when Assumptions A1 and A2 are not satisfied.

**Importance of the KL-divergence Regularization.** Recall that the adaptation objectives Eqs. (6) and (17) include the KL-divergence term scaled by $\lambda$ in order to avoid large distribution changes when there is not enough support from the small target-domain dataset. A natural question to ask is whether this term is useful and helps improve the adaptation solution when $\lambda > 0$. To answer this, we compare the performance of our method with $\lambda = 0$ with that our our method with $\lambda$ set automatically using the validation metric. The results of this comparison are given in Fig. 9 on four simulated channel variations. The results are averaged over multiple trials as before. It is clear that setting $\lambda = 0$ for our method leads to much higher symbol error rates compared to setting $\lambda$ to a non-zero value using the validation metric, establishing the importance of the KL-divergence term.

**Performance Under No Distribution Change.** We evaluate the symbol error rate performance of all the methods in the setting where there is no distribution change. In this setting, the performance of the MDN and autoencoder should not change, and we expect the proposed adaptation method to maintain a similar performance (not lead to increased symbol error rate). In Fig. 10, we report the results of this experiment when both the source and target channel distributions are either Ricean fading or Uniform fading. We consider a medium SNR value of 14 dB and a high SNR value of 20 dB. We observe that our method is relatively stable even when there is no distribution change, and there is only a small increase in error rate. For instance, in Fig. 10c, the error rate of our method increases from 0.015 to 0.018 for 5 samples per class.

We expect that a practical system that frequently adapts to changes in the channel distribution should first have a distribution change-detection algorithm that takes a batch of new samples from the channel and detects whether there is any change in the distribution. The actual domain adaptation algorithm is then applied only when a distribution change is detected. In this way, any potential drop in the autoencoder's performance when there is no distribution change can be made less likely.

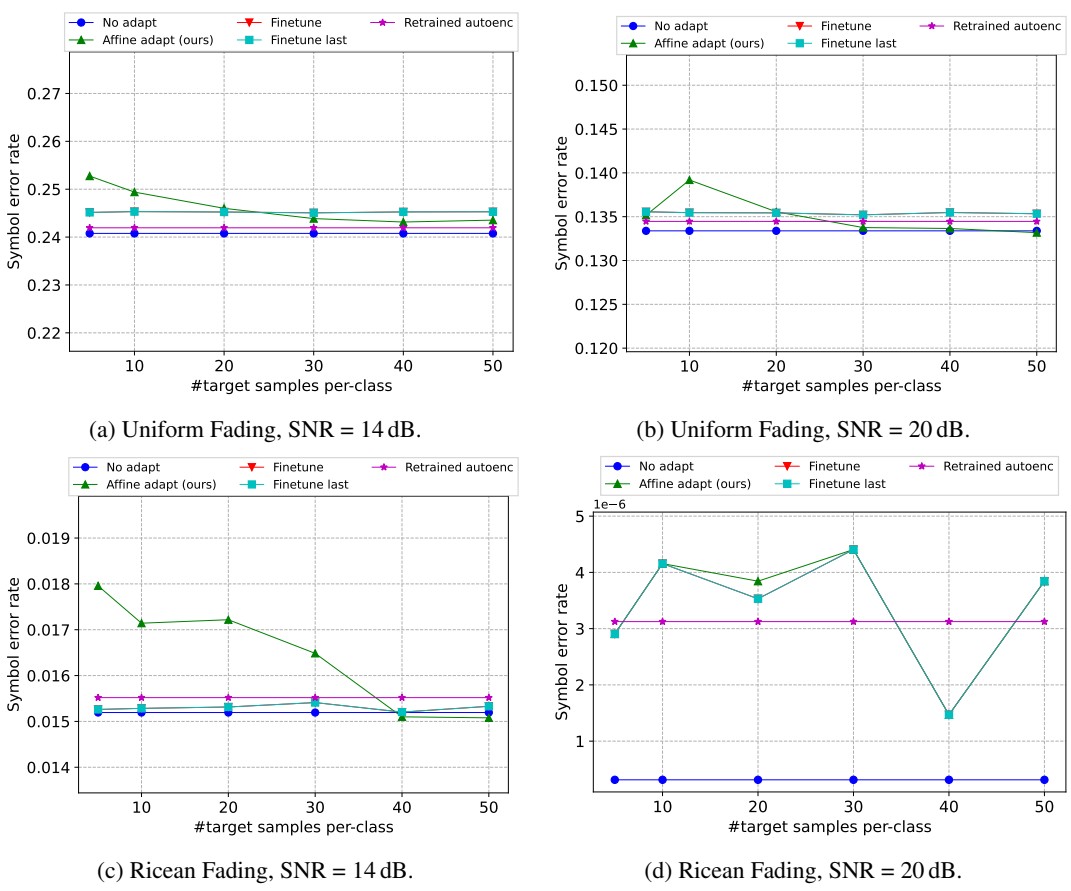

Figure 10: Performance under no distribution change, *i.e.*, the source and target distributions are the same. The type of channel distribution and the corresponding SNR common to both the source and target domain is indicated. In Fig. (d) note that the y-axis is scaled by $10^6$.

## C.5 Analysis of the Failure on AWGN to Ricean Fading

Referring to Fig. 4. c in the main paper, we observe that our method has a worse symbol error rate compared to no adaptation and the other baselines for the adaptation setting from an AWGN channel at 14dB SNR to a Ricean fading channel at 20dB SNR. In order to get an intuition for this, we look into the data distribution plots from the source and target channels (see Fig. 12). From the figure, we observe that the target distribution is actually very class-separable, but the distribution shift between

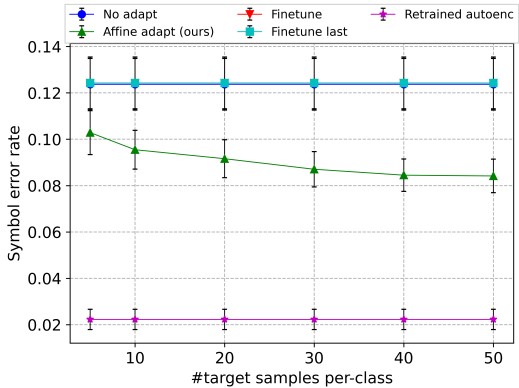

Figure 11: Performance under distribution shift where the source and target distributions are random Gaussian mixtures with mismatched components.

the AWGN and Ricean fading channels is quite large, particularly because of the high SNR of the Ricean fading channel. In this case, the classifier from the source domain is able to classify the target domain data reasonably well without any adaptation. The fine-tuning baselines ("Finetune" and "Finetune last") do not lead to much improvement and have similar SER as the case of no adaptation.

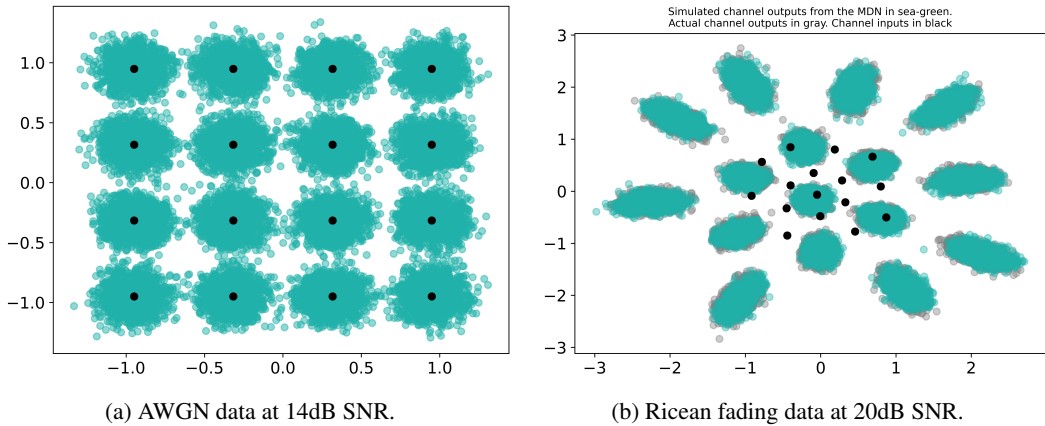

(a) AWGN data at 14dB SNR.

(b) Ricean fading data at 20dB SNR.

Figure 12: Data plots of the source domain (AWGN at 14dB SNR) and the target domain (Ricean fading at 20dB SNR). The black points are the channel inputs (*i.e.*, the symbols). A 16-QAM constellation is used in the source domain (left), while the learned autoencoder constellation is used in the target domain (right).

On the other hand, the proposed method leads to worse SER on the target domain after adaptation. We hypothesize that it fails in this case because the set of affine transformation are unable to capture the large distribution change well. In our current formulation, the affine transformations per component are shared by all the classes. From Eq. 4, the affine transformation parameters are $\mathbf{A}_i, \mathbf{b}_i, \mathbf{C}_i, \beta_i$, and $\gamma_i, \ i \in [k]$. Based on the data plots, we think that allowing the affine transformations to be class-specific can handle this adaptation setting better. While allowing the affine transformations to be class-specific provides more flexibility, it also introduces more parameters which need to be optimized using a small dataset from the target distribution.

## C.6 EXPERIMENTS ON GENERATIVE ADAPTATION OF THE MDN

We evaluate the generative adaptation method of the MDN discussed in Appendix B.3, where the goal is to achieve a high conditional log-likelihood (CLL) under the target data distribution. As shown in Fig. 13, we consider the standard simulated distribution changes including random Gaussian mixtures, and vary the number of target samples per-class from 2 to 20. For each pair of source-target distributions and each target sample size, the experiment is repeated for 50 trials, and the average CLL

is reported. The CLL is calculated on a held-out test dataset of size 25,000 from the target distribution. For the simulated channels, the SNR of the source and target distributions are each randomly chosen from the range 10 dB to 20 dB. From Fig. 13, we observe that the proposed MDN adaptation has significantly higher CLL compared to no adaptation and the finetuning baseline methods, especially for very small number of target samples per class.

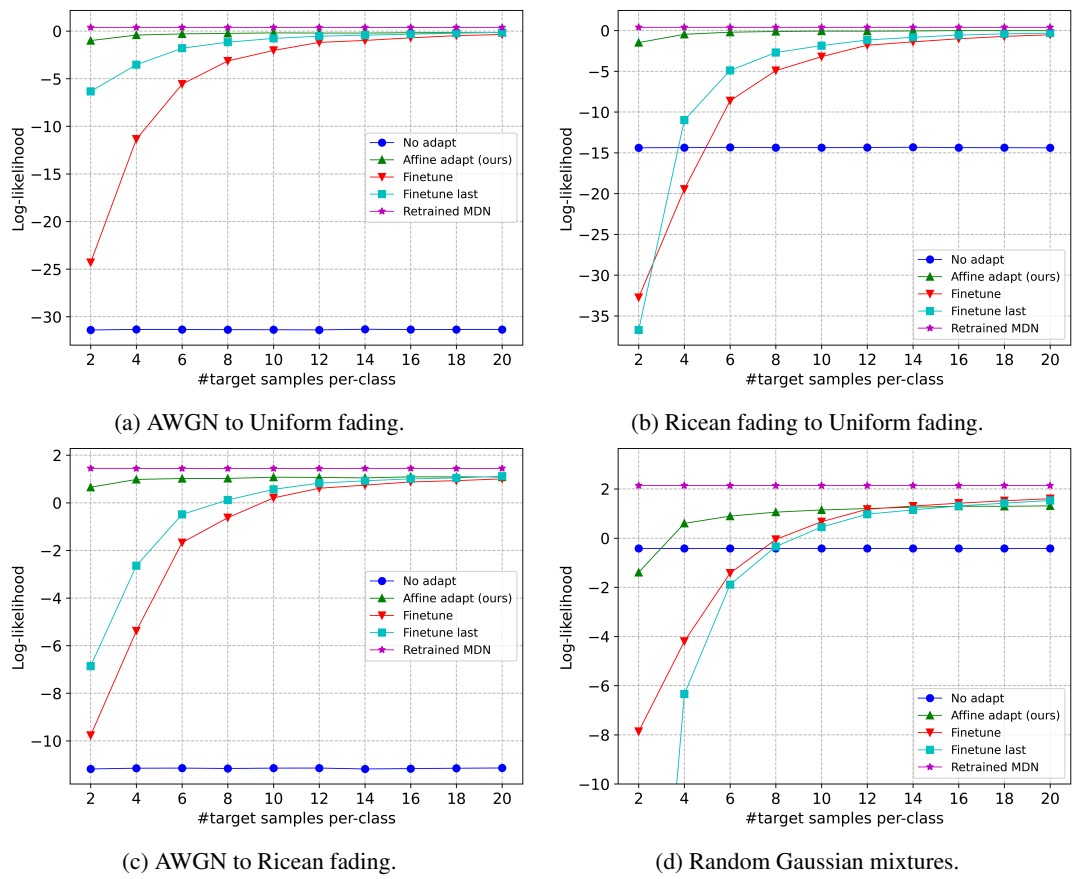

(a) AWGN to Uniform fading.

(b) Ricean fading to Uniform fading.

(c) AWGN to Ricean fading.

(d) Random Gaussian mixtures.

Figure 13: Generative adaptation of the MDN channel to different types of distribution shifts. Larger values of conditional log-likelihood correspond to better adaptation solutions.

## C.7 EVALUATION WITH VERY FEW TARGET DOMAIN SAMPLES

To evaluate how the proposed method performs with fewer than 10 samples per class, we ran an experiment on some simulated distribution changes with $1, 2, 4, 6,$ and $8$ samples per class. We report these results in Fig. 14 for the following distribution changes: 1) Random Gaussian mixtures (Fig. 14a), and 2) Ricean fading to Uniform fading (Fig. 14b). We follow the same protocol as in the main paper and report the average results from multiple random trials. From Fig. 14a, corresponding to random Gaussian mixtures, we observe that the proposed method is able to improve performance (decrease SER) starting from 1 sample per class.

However, in Fig. 14a (Ricean fading to Uniform fading), the proposed method increases the SER (worse) compared to no adaptation for 1 and 2 samples per class. The SER starts to decrease significantly from 4 samples per class. We hypothesize that the reason for failure of the proposed method with 1 and 2 samples per class could be the strong distribution change and complexity of the target uniform fading channel. While this may seem concerning, please note that we chose these strong simulated distribution changes in order to demonstrate the potential improvements of our method. In practical wireless channels, the distribution changes are likely to be more gradual, so that the proposed method can usually adapt well with only a few samples per class.

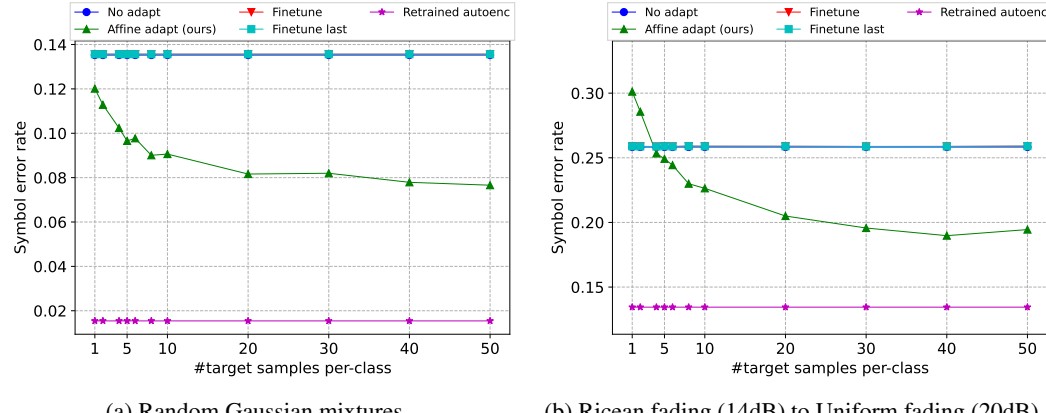

(a) Random Gaussian mixtures.       (b) Ricean fading (14dB) to Uniform fading (20dB).

Figure 14: Results of autoencoder adaptation on simulated distribution changes with very few target domain samples (starting from 1 sample per class). For the case where both the source and target domain data are from random Gaussian mixtures, our method is able to improve the SER even with 1 sample per class. In the case of adaptation from Ricean fading to Uniform fading, our method increases the SER (compared to no adaptation) for 1 and 2 samples per class, but starts to decrease the SER after that.

## C.8   GAP BETWEEN THE PROPOSED METHOD AND THE RETRAINED AUTOENCODER

In this section, we address the following question: Does the performance of the proposed method approach that of the oracle retrained method as the size of the adaptation dataset increases? We note that the oracle method "Retrained autoenc" retrains the channel model, encoder and decoder networks jointly using Algorithm 1 in Appendix D.3. In our experiments, we used $20{,}000$ samples from the target (domain) distribution for training the MDN channel in between each epoch of the encoder/decoder update, and used $300{,}000$ samples for training the encoder/decoder networks with the MDN parameters fixed. The retrained autoencoder therefore learns a custom *optimal* constellation (encoding) for the target distribution.

Table 4: Performance of the proposed adaptation method with a large number of target domain samples, and comparison with an oracle retrained autoencoder for the target domain.

| Source and target domains | Proposed method (50 samples/class) | Proposed method (1000 samples/class) | Retrained autoencoder |
|---|---|---|---|
| Random Gaussian mixtures | 0.0766 | 0.0695 | 0.0154 |
| AWGN to Uniform fading | 0.2370 | 0.2138 | 0.1345 |
| Ricean fading to Uniform fading | 0.1945 | 0.1815 | 0.1345 |
| AWGN to Ricean fading | 6.381e-4 | 2.531e-5 | 3.125e-6 |
| Uniform fading to Ricean fading | 6.484e-4 | 5.503e-4 | 3.125e-6 |

Our adaptation method, on the other hand, does not modify the constellation learned by the autoencoder from the source domain data, which could be sub-optimal for the target domain. This restriction of not modifying the constellation is practically advantageous since there is no need to communicate the new symbols back to the transmitter. Also, there is no need to change the transmitter side encoding frequently. Since the proposed method does not have the full flexibility of the retrained autoencoder, we do not expect its performance (SER) to converge to that of the latter under significant distribution change. For instance, Fig. 12b shows the learned autoencoder constellation for a Ricean fading channel of 20 dB SNR. This is quite different from the optimal constellation for an AWGN channel of 14 dB SNR (which is closer to a 16-QAM constellation).

To understand this better, we ran some experiments to evaluate our method with a large (target) adaptation dataset, and compared it to the oracle retrained autoencoder. In Table 4, we report the SER of our method with $1000$ samples per class (so $16{,}000$ samples overall) on a subset of simulated

distribution changes. We follow the same protocol as the main paper, and report the average SER from multiple random trials on a large test set from the target distribution.

We also report the performance of our method with 50 samples/class for comparison. From the table, we observe that there is a gap in the SER of our method (with 1000 samples/class) and that of the retrained autoencoder. We believe that allowing the encoder symbols to be optimized might be required in order to bridge this gap in SER. In conclusion, when it is possible to collect sufficient (large) data from the target distribution, it might be better to allow the encoder network to also be optimized via retraining.

## D DETAILED BACKGROUND

Expanding on § 2, we provide additional background on the following topics: 1) components of an end-to-end autoencoder-based communication system, 2) generative modeling using mixture density networks, 3) training algorithm of the autoencoder, and 4) a primer on domain adaptation.

### D.1 AUTOENCODER-BASED END-TO-END LEARNING

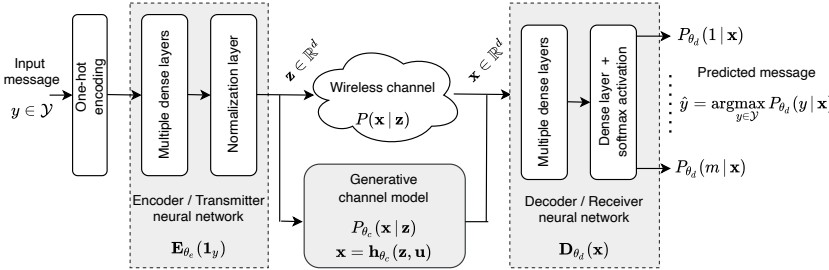

Figure 15: An end-to-end autoencoder-based communication system with a generative channel model.

Consider a single-input, single-output wireless communication system as shown in Fig. 1, where the transmitter encodes and transmits messages $y$ from the set $\mathcal{Y} = \{1, 2, \cdots, m\}$ to the receiver through $d \geq 2$ discrete uses of the wireless channel. The receiver attempts to accurately decode the transmitted message from the distorted and noisy channel output $\mathbf{x}$. We discuss the end-to-end learning of such a system using the concept of autoencoders (O'Shea & Hoydis, 2017; Dörner et al., 2018).

**Transmitter / Encoder Neural Network.** The transmitter or encoder part of the autoencoder is modeled as a multi-layer, feed-forward neural network (NN) that takes as input the one-hot-coded representation $\mathbf{1}_y$ of a message $y \in \mathcal{Y}$, and produces an encoded symbol vector $\mathbf{z} = \mathbf{E}_{\boldsymbol{\theta}_e}(\mathbf{1}_y) \in \mathbb{R}^d$. Here, $\boldsymbol{\theta}_e$ are the parameters (weights and biases) of the encoder NN and $d$ is the encoding dimension. Due to hardware constraints present at the transmitter, a normalization layer is used as the final layer of the encoder network in order to constrain the average power and/or the amplitude of the symbol vectors. The average power constraint is defined as $\mathbb{E}[\|\mathbf{z}\|_2^2] = \mathbb{E}_y[\|\mathbf{E}_{\boldsymbol{\theta}_e}(\mathbf{1}_y)\|_2^2] \leq c$, where the expectation is over the prior distribution of the input messages, and $c$ is typically set to 1. The amplitude constraint is defined as $|z_i| \leq 1, \forall i \in [d]$. The size of the message set is usually chosen to be a power of 2, *i.e.*, $m = 2^b$ representing $b$ bits of information. Following O'Shea & Hoydis (2017), the communication rate of this system is the number of bits transmitted per channel use, which in this case is $R = b / d$. An autoencoder transmitting $b$ bits over $d$ uses of the channel is referred to as a $(d, b)$ autoencoder. For example, a $(2, 4)$ autoencoder uses a message set of size 16 and an encoding dimension of 2, with a communication rate $R = 2$ bits/channel use.

**Receiver / Decoder Neural Network.** The receiver or decoder component is also a multilayer, feedforward NN that takes the channel output $\mathbf{x} \in \mathbb{R}^d$ as its input and outputs a probability distribution over the $m$ messages. The input-output mapping of the decoder NN can be expressed as $\mathbf{D}_{\boldsymbol{\theta}_d}(\mathbf{x}) := [P_{\boldsymbol{\theta}_d}(1 \,|\, \mathbf{x}), \cdots, P_{\boldsymbol{\theta}_d}(m \,|\, \mathbf{x})]^T$, where $\boldsymbol{\theta}_d$ are the parameters of the decoder NN. The softmax activation function is used at the final layer to ensure that the outputs are valid probabilities. The message corresponding to the highest output probability is predicted as the decoded message, *i.e.*, $\widehat{y}(\mathbf{x}) = \mathrm{argmax}_{y \in \mathcal{Y}} \; P_{\boldsymbol{\theta}_d}(y \,|\, \mathbf{x})$. The decoder NN is essentially a *discriminative classifier* that

learns to accurately categorize the received (distorted) symbol vector into one of the $m$ message classes. This is in contrast to conventional autoencoders, where the decoder learns to accurately reconstruct a high-dimensional tensor input from its low-dimensional representation learned by the encoder. In the case of communication autoencoders, the *symbol or block error rate (BLER)*, defined as $\mathbb{E}_{(\mathbf{x},y)}[\mathbb{1}(\widehat{y}(\mathbf{x}) \neq y)]$, is used as the end-to-end performance metric.

**Channel Model.** As discussed in § 1, the wireless channel can be represented by a *conditional probability density* of the channel output given its input $p(\mathbf{x} \mid \mathbf{z})$. The channel can be equivalently characterized by a *stochastic transfer function* $\mathbf{x} = \mathbf{h}(\mathbf{z}, \mathbf{u})$ that transforms the encoded symbol vector into the channel output, where $\mathbf{u}$ captures the stochastic components of the channel (*e.g.*, random noise, phase offsets). For example, an additive white Gaussian noise (AWGN) channel is represented by $\mathbf{x} = \mathbf{h}(\mathbf{z}, \mathbf{u}) = \mathbf{z} + \mathbf{u}$, with $\mathbf{u} \sim \mathcal{N}(\cdot \mid \mathbf{0}, \sigma^2 \mathbf{I}_d)$ and $p(\mathbf{x} \mid \mathbf{z}) = \mathcal{N}(\mathbf{x} \mid \mathbf{z}, \sigma^2 \mathbf{I}_d)$. For realistic wireless channels, the transfer function and conditional probability density are usually unknown and hard to approximate well with standard mathematical models. Recently, a number of works have applied generative models such as conditional generative adversarial networks (GANs) (O'Shea et al., 2019; Ye et al., 2018), MDNs (García Martí et al., 2020), and conditional variational autoencoders (VAEs) (Xia et al., 2020) for modeling the wireless channel. To model a wireless channel, generative methods learn a parametric model $P_{\boldsymbol{\theta}_c}(\mathbf{x} \mid \mathbf{z})$ (possibly a neural network) that closely approximates the true conditional density of the channel from a dataset of channel input/output observations. Learning a generative model of the channel comes with *important advantages*. 1) Once the parameters of the channel model are learned from data, the model can be used to generate *any number of representative samples* from the channel distribution. 2) A channel model with a differentiable transfer function makes it possible to backpropagate gradients of the autoencoder loss through the channel and train the autoencoder using stochastic gradient descent (SGD)-based optimization. 3) It allows for continuous adaptation of the generative channel model to variations in the channel conditions, and thereby maintain a low error rate of the autoencoder.

## D.2 GENERATIVE CHANNEL MODEL USING A MIXTURE DENSITY NETWORK

As discussed in § 2, we use an MDN (Bishop, 1994; 2007) with Gaussian components to model the conditional density of the channel. MDNs can model complex conditional densities by combining a (feed-forward) neural network with a standard parametric mixture model (*e.g.*, mixture of Gaussians). The MDN learns to predict the parameters of the mixture model $\boldsymbol{\phi}(\mathbf{z})$ as a function of the channel input $\mathbf{z}$. This can be expressed as $\boldsymbol{\phi}(\mathbf{z}) = \mathbf{M}_{\boldsymbol{\theta}_c}(\mathbf{z})$, where $\boldsymbol{\theta}_c$ are the parameters of the neural network. The parameters of the mixture model defined by the MDN are a concatenation of the parameters from the $k$ density components, *i.e.*, $\boldsymbol{\phi}(\mathbf{z})^T = [\boldsymbol{\phi}_1(\mathbf{z})^T, \cdots, \boldsymbol{\phi}_k(\mathbf{z})^T]$, where $\boldsymbol{\phi}_i(\mathbf{z})$ is the parameter vector of component $i$. Focusing on a Gaussian mixture, the *channel conditional density* given each symbol $\mathbf{z} \in \mathcal{Z}$ is given by

$$P_{\boldsymbol{\theta}_c}(\mathbf{x} \mid \mathbf{z}) = \sum_{i=1}^{k} P_{\boldsymbol{\theta}_c}(K = i \mid \mathbf{z}) \, P_{\boldsymbol{\theta}_c}(\mathbf{x} \mid \mathbf{z}, K = i) = \sum_{i=1}^{k} \pi_i(\mathbf{z}) \, N\big(\mathbf{x} \mid \boldsymbol{\mu}_i(\mathbf{z}), \boldsymbol{\Sigma}_i(\mathbf{z})\big), \quad (20)$$

where $\boldsymbol{\mu}_i(\mathbf{z}) \in \mathbb{R}^d$ is the mean vector, $\boldsymbol{\Sigma}_i(\mathbf{z}) \in \mathbb{R}^{d \times d}$ is the (symmetric, positive-definite) covariance matrix, and $\pi_i(\mathbf{z}) \in [0, 1]$ is the weight (prior probability) of component $i$. Also, $K$ is the latent random variable denoting the mixture component of origin. The weights of the mixture are parameterized using the softmax function as $\pi_i(\mathbf{z}) = e^{\alpha_i(\mathbf{z})} / \sum_{j=1}^{k} e^{\alpha_j(\mathbf{z})}, \; \forall i$ in order to satisfy the probability constraint. The MDN simply predicts the un-normalized weights $\alpha_i(\mathbf{z}) \in \mathbb{R}$ (also known as the *prior logits*). We define the parameter vector of component $i$ as $\boldsymbol{\phi}_i(\mathbf{z})^T = [\alpha_i(\mathbf{z}), \boldsymbol{\mu}_i(\mathbf{z})^T, \text{vec}(\boldsymbol{\Sigma}_i(\mathbf{z}))^T]$, where $\text{vec}(\cdot)$ is the vector representation of the unique entries of the covariance matrix. Details on the *conditional log-likelihood (CLL)* training objective and the *transfer function* of the MDN can be found in Appendix E.

## D.3 TRAINING OF THE AUTOENCODER

In this section, we provide a formal discussion of the end-to-end training of the autoencoder. First, we define the input-output mapping of the autoencoder as $\mathbf{f}_{\boldsymbol{\theta}}(\mathbf{1}_y) = \mathbf{D}_{\boldsymbol{\theta}_d}(\mathbf{h}_{\boldsymbol{\theta}_c}(\mathbf{E}_{\boldsymbol{\theta}_e}(\mathbf{1}_y), \mathbf{u})) = (\mathbf{D}_{\boldsymbol{\theta}_d} \circ \mathbf{h}_{\boldsymbol{\theta}_c}(\cdot, \mathbf{u}) \circ \mathbf{E}_{\boldsymbol{\theta}_e})(\mathbf{1}_y)$, where $\boldsymbol{\theta}^T = [\boldsymbol{\theta}_e^T, \boldsymbol{\theta}_c^T, \boldsymbol{\theta}_d^T]$ is the combined vector of parameters from the encoder, channel, and decoder. Given an input message $y \in \mathcal{Y}$, the autoencoder maps the one-hot-coded representation of $y$ into an output probability vector over the message set. Note that, while the

encoder and decoder neural networks are deterministic, a forward pass through the autoencoder is stochastic due to the channel transfer function $\mathbf{h}_{\boldsymbol{\theta}_c}$. The learning objective of the autoencoder is to accurately recover the input message at the decoder with a high probability. The cross-entropy (CE) loss, which is commonly used for training classifiers, is also suitable for end-to-end training of the autoencoder. For an input $y$ with encoded representation $\mathbf{z} = \mathbf{E}_{\boldsymbol{\theta}_e}(\mathbf{1}_y)$, channel output $\mathbf{x} = \mathbf{h}_{\boldsymbol{\theta}_c}(\mathbf{z}, \mathbf{u})$, and decoded output $\mathbf{D}_{\boldsymbol{\theta}_d}(\mathbf{x}) = [P_{\boldsymbol{\theta}_d}(1 \mid \mathbf{x}), \cdots, P_{\boldsymbol{\theta}_d}(m \mid \mathbf{x})]^T$, the CE loss is given by

$$
\begin{aligned}
\ell_{\mathrm{CE}}(\mathbf{1}_y, \mathbf{f}_{\boldsymbol{\theta}}(\mathbf{1}_y)) \;=\; -\mathbf{1}_y^T \log \mathbf{f}_{\boldsymbol{\theta}}(\mathbf{1}_y) \;&=\; -\mathbf{1}_y^T \log \mathbf{D}_{\boldsymbol{\theta}_d}\big(\mathbf{h}_{\boldsymbol{\theta}_c}(\mathbf{E}_{\boldsymbol{\theta}_e}(\mathbf{1}_y)), \mathbf{u})\big) \\
&=\; -\log P_{\boldsymbol{\theta}_d}\big(y \mid \mathbf{h}_{\boldsymbol{\theta}_c}(\mathbf{E}_{\boldsymbol{\theta}_e}(\mathbf{1}_y)), \mathbf{u})\big),
\end{aligned}
\tag{21}
$$

which is always non-negative and takes the minimum value 0 when the correct message is decoded with probability 1. The autoencoder aims to minimize the following expected CE loss over the input message set and the channel output:

$$
\mathbb{E}[\ell_{\mathrm{CE}}(\mathbf{1}_y, \mathbf{f}_{\boldsymbol{\theta}}(\mathbf{1}_y))] \;=\; -\sum_{y=1}^m p(y) \int_{\mathbb{R}^d} P_{\boldsymbol{\theta}_c}(\mathbf{x} \mid \mathbf{E}_{\boldsymbol{\theta}_e}(\mathbf{1}_y)) \, \log P_{\boldsymbol{\theta}_d}(y \mid \mathbf{x}) \, d\mathbf{x}.
\tag{22}
$$

Here $\{p(y), \; \forall y \in \mathcal{Y}\}$ is the prior probability over the input messages, which is usually taken to be uniform in the absence of prior knowledge. In practice, the autoencoder minimizes an empirical estimate of the expected CE loss function by generating a large set of samples from the channel conditional density given each message. Let $\mathcal{X}^y = \{\mathbf{x}_n^y = \mathbf{h}_{\boldsymbol{\theta}_c}(\mathbf{E}_{\boldsymbol{\theta}_e}(\mathbf{1}_y), \mathbf{u}_n), \; n = 1, \cdots, N\}$ denote a set of independent and identically distributed (iid) samples from $P_{\boldsymbol{\theta}_c}(\mathbf{x} \mid \mathbf{E}_{\boldsymbol{\theta}_e}(\mathbf{1}_y))$, the channel conditional density given message $y$. Also, let $\mathcal{X} = \cup_y \mathcal{X}^y$ denote the combined set of samples from all messages. The empirical expectation of the autoencoder CE loss (22) is given by

$$
\mathcal{L}_{auto}(\boldsymbol{\theta} \,; \mathcal{X}) \;=\; -\sum_{y=1}^m p(y) \frac{1}{N} \sum_{n=1}^N \log P_{\boldsymbol{\theta}_d}\big(y \mid \mathbf{h}_{\boldsymbol{\theta}_c}(\mathbf{E}_{\boldsymbol{\theta}_e}(\mathbf{1}_y), \mathbf{u}_n)\big).
\tag{23}
$$

It is clear from the above equation that the channel transfer function $\mathbf{h}_{\boldsymbol{\theta}_c}$ should be differentiable in order to be able to backpropagate gradients through the channel to the encoder network. The transfer function defining sample generation for a Gaussian MDN channel is discussed in Appendix E.

---

**Algorithm 1** End-to-end training of the autoencoder with an MDN channel

---

1: **Inputs:** Message size $m$; Encoding dimension $d$; Initial constellation $\{\mathbf{E}_0(\mathbf{1}_y), \; \forall y \in \mathcal{Y}\}$; Number of optimization epochs for the autoencoder $N_{ae}$ and channel $N_{ce}$.
2: **Output:** Trained network parameters $\boldsymbol{\theta}_e, \boldsymbol{\theta}_c, \boldsymbol{\theta}_d$.

3: Initialize the encoder, channel, and decoder network parameters.
4: Sample training data $\mathcal{D}_c^{(0)}$ from the channel using the initial constellation.
5: Train the channel model for $N_{ce}$ epochs to minimize $\mathcal{L}_{\mathrm{ch}}(\boldsymbol{\theta}_c \,; \mathcal{D}_c^{(0)})$ (Eq. 24).
6: **for** epoch $t = 1, \cdots, N_{ae}$:
7:     Freeze the channel model parameters $\boldsymbol{\theta}_c$.
8:     Perform a round of mini-batch SGD updates of $\boldsymbol{\theta}_e$ and $\boldsymbol{\theta}_d$ with respect to $\mathcal{L}_{auto}(\boldsymbol{\theta} \,; \mathcal{X})$.
9:     Sample data $\mathcal{D}_c^{(t)}$ from the channel using the updated constellation $\{\mathbf{E}_{\boldsymbol{\theta}_e}(\mathbf{1}_y), \; \forall y \in \mathcal{Y}\}$.
10:     Train the channel model for $N_{ce}$ epochs to minimize $\mathcal{L}_{\mathrm{ch}}(\boldsymbol{\theta}_c \,; \mathcal{D}_c^{(t)})$ (Eq. 24).
11: **end for**

12: **Return** $\boldsymbol{\theta}_e, \boldsymbol{\theta}_c, \boldsymbol{\theta}_d$.

---

The training algorithm for jointly learning the autoencoder and channel model (*e.g.*, García Martí et al. (2020)) is given in Algorithm 1. It is an alternating (cyclic) optimization of the channel parameters and the autoencoder (encoder and decoder) parameters. The reason this type of alternating optimization is required is because the empirical expectation of the CE loss Eq. (23) is valid *only when* the channel conditional density (*i.e.*, $\boldsymbol{\theta}_c$) is fixed. The training algorithm can be summarized as follows. First, the channel model is trained for $N_{ce}$ epochs using data sampled from the channel with an initial encoder constellation (*e.g.*, M-QAM). With the channel model parameters fixed, the parameters of the encoder and decoder networks are optimized for one epoch of mini-batch SGD updates (using any adaptive learning rate algorithm *e.g.*, Adam (Kingma & Ba, 2015)). Since the channel model is no longer

optimal for the updated encoder constellation, it is retrained for $N_{ce}$ epochs using data sampled from the channel with the updated constellation. This alternate training of the encoder/decoder and the channel networks is repeated for $N_{ae}$ epochs or until convergence.

Finally, we observe some interesting nuances of the communication autoencoder learning task that is not common to other domains such as images. 1) The size of the input space is finite, equal to the number of distinct messages $m$. Because of the stochastic nature of the channel transfer function, the same input message results in a different autoencoder output each time. 2) There is theoretically no limit on the number of samples that can be generated for training and validating the autoencoder. These factors make the autoencoder learning less susceptible to overfitting, unlike in other domains.

### D.4 A PRIMER ON DOMAIN ADAPTATION

We provide a brief review of domain adaptation (DA) problem and literature. In the traditional learning setting, training and test data are assumed to be sampled independently from the same distribution $P(\mathbf{x}, y)$, where $\mathbf{x}$ and $y$ are the input vector and target respectively [8]. In many real world settings, it can be hard or impractical to collect a large labeled dataset $\mathcal{D}_t^\ell$ for a *target domain* where the machine learning model (*e.g.*, a DNN classifier) is to be deployed. On the other hand, it is common to have access to a large unlabeled dataset $\mathcal{D}_t^u$ from the target domain, and a large labeled dataset $\mathcal{D}_s^\ell$ from a different but related *source domain* [9]. Both $\mathcal{D}_s^\ell$ and $\mathcal{D}_t^u$ are much larger than $\mathcal{D}_t^\ell$, and in most cases there is no labeled data from the target domain (referred to as unsupervised DA). For the target domain, the unlabeled dataset (and labeled dataset if any) are sampled from an unknown target distribution, *i.e.*, $\mathbf{x} \in \mathcal{D}_t^u \sim P_t(\mathbf{x})$ and $(\mathbf{x}, y) \in \mathcal{D}_t^\ell \sim P_t(\mathbf{x}, y)$. For the source domain, the labeled dataset is sampled from an unknown source distribution, *i.e.*, $(\mathbf{x}, y) \in \mathcal{D}_s^\ell \sim P_s(\mathbf{x}, y)$. The goal of unsupervised DA is to leverage the available labeled and unlabeled datasets from the two domains to learn a predictor, denoted by the parametric function $\hat{y} = f_{\boldsymbol{\theta}}(\mathbf{x})$, such that the following risk function w.r.t the target distribution is minimized:

$$R_t[f_{\boldsymbol{\theta}}] = \mathbb{E}_{(\mathbf{x}, y) \sim P_t}[\ell(f_{\boldsymbol{\theta}}(\mathbf{x}), y)] = \sum_y \int_{\mathbf{x}} P_t(\mathbf{x}, y)\, \ell(f_{\boldsymbol{\theta}}(\mathbf{x}), y)\, d\mathbf{x},$$

where $\ell(\hat{y}, y)$ is a loss function that penalizes the prediction $\hat{y}$ for deviating from the true value $y$ (*e.g.*, cross-entropy or hinge loss). In a similar way, we can define the risk function w.r.t the source distribution $R_s[f_{\boldsymbol{\theta}}]$. A number of seminal works in DA theory (Ben-David et al., 2006; Blitzer et al., 2007; Ben-David et al., 2010) have studied this learning setting and provide bounds on $R_t[f_{\boldsymbol{\theta}}]$ in terms of $R_s[f_{\boldsymbol{\theta}}]$ and the divergence between source and target domain distributions. Motivated by this foundational theory, a number of recent works (Ganin & Lempitsky, 2015; Ganin et al., 2016; Long et al., 2018; Saito et al., 2018; Zhao et al., 2019; Johansson et al., 2019) have proposed using DNNs for adversarially learning a shared representation across the source and target domains such that a predictor using this representation and trained using labeled data from only the source domain also generalizes well to the target domain. An influential work in this line of DA is the domain adversarial neural network (DANN) proposed by Ganin & Lempitsky (2015) and later by Ganin et al. (2016). The key idea behind the DANN approach is to adversarially train a label predictor NN and a domain discriminator NN in order to learn a feature representation for which i) the source and target inputs are nearly indistinguishable to the domain discriminator, and ii) the label predictor has good generalization performance on the source domain inputs.

**Special Cases of DA.** While the general DA problem addresses the scenario where $P_s(\mathbf{x}, y)$ and $P_t(\mathbf{x}, y)$ are different, certain special cases of DA have also been explored. One such special case is *covariate shift* (Sugiyama et al., 2007; Sugiyama & Kawanabe, 2012), where only the marginal distribution of the inputs changes (*i.e.*, $P_t(\mathbf{x}) \neq P_s(\mathbf{x})$), but the conditional distribution of the target given the input does not change (*i.e.*, $P_t(y \mid \mathbf{x}) \approx P_s(y \mid \mathbf{x})$). Another special case is the so-called *label shift* or class-prior mismatch (Saerens et al., 2002; Du Plessis & Sugiyama, 2014), where only the marginal distribution of the label changes (*i.e.*, $P_t(y) \neq P_s(y)$), but the conditional distribution of the input given the target does not change (*i.e.*, $P_t(\mathbf{x} \mid y) \approx P_s(\mathbf{x} \mid y)$). Prior works have proposed targeted theory and methods for these special cases of DA.

---

[8] The notation used in this section is different from the rest of the paper, but consistent with the statistical learning literature.

[9] One could have multiple source domains in practice; we consider the single source domain setting.

# E  MDN TRAINING AND SAMPLE GENERATION

In § 2, we briefly discussed how a Gaussian mixture density network (MDN) can be used to learn a generative model for the channel. Here, we provide details on the training algorithm for the MDN, followed by a discussion on the sampling function $\mathbf{h}_{\boldsymbol{\theta}_c}$ of an MDN, and how to make the sampling function differentiable to enable SGD-based training of the autoencoder. Given a dataset of input-output pairs sampled from the channel $\mathcal{D}_c = \{(\mathbf{z}_n, \mathbf{x}_n),\ n = 1, \cdots, N_c\}$, the MDN is trained to minimize the negative conditional log-likelihood (CLL) of the data given by

$$\mathcal{L}_{\mathrm{ch}}(\boldsymbol{\theta}_c\,;\mathcal{D}_c) \;=\; -\frac{1}{N_c} \sum_{n=1}^{N_c} \log P_{\boldsymbol{\theta}_c}(\mathbf{x}_n \,|\, \mathbf{z}_n), \tag{24}$$

where the Gaussian mixture $P_{\boldsymbol{\theta}_c}(\mathbf{x}_n \,|\, \mathbf{z}_n)$ is given by Eq. (1). With a large $N_c$, the MDN can learn a sufficiently-complex parametric density model of the channel. The negative CLL objective can be interpreted as the sample estimate of the Kullback-Leibler divergence between the true (unknown) conditional density $P(\mathbf{x} \,|\, \mathbf{z})$ and the conditional density modeled by the MDN $P_{\boldsymbol{\theta}_c}(\mathbf{x} \,|\, \mathbf{z})$. Therefore, minimizing the negative CLL finds the MDN parameters $\boldsymbol{\theta}_c$ that lead to a close approximation of the true conditional density. Standard SGD-based optimization methods such as Adam (Kingma & Ba, 2015) can be applied to find the MDN parameters $\boldsymbol{\theta}_c$ that (locally) minimize the negative CLL.

After the MDN is trained, new simulated samples from the channel distribution can be generated from the Gaussian mixture using the following stochastic sampling method. We focus on the diagonal covariance case for simplicity, where $\boldsymbol{\sigma}_i^2(\mathbf{z}) \in \mathbb{R}_+^d$ are the diagonal elements of the covariance matrix $\boldsymbol{\Sigma}_i(\mathbf{z})$ for component $i$.

1. Randomly select an encoded symbol $\mathbf{z}$ from the constellation according to the prior distribution $\{p(\mathbf{z}),\ \mathbf{z} \in \mathcal{Z}\}$ [10].
2. Randomly select a component $K = i$ according to the mixture weights $\{\pi_1(\mathbf{z}), \cdots, \pi_k(\mathbf{z})\}$.
3. Randomly sample $\mathbf{u}$ from the isotropic $d$-dimensional Gaussian density $\mathbf{u} \sim N(\cdot \,|\, \mathbf{0}, \mathbf{I}_d)$.
4. Generate the channel output as $\mathbf{x} \;=\; \boldsymbol{\sigma}_i^2(\mathbf{z}) \odot \mathbf{u} \;+\; \boldsymbol{\mu}_i(\mathbf{z})$.

Here $\odot$ refers to the element-wise (Hadamard) product of two vectors. The channel transfer or sampling function for a Gaussian MDN can thus be expressed as

$$\mathbf{x} \;=\; \mathbf{h}_{\boldsymbol{\theta}_c}(\mathbf{z}, \mathbf{u}) \;=\; \sum_{i=1}^{k} \mathbb{1}(K = i)\,(\boldsymbol{\sigma}_i^2(\mathbf{z}) \odot \mathbf{u} \;+\; \boldsymbol{\mu}_i(\mathbf{z})), \tag{25}$$

where $K \sim \mathrm{Cat}(\pi_1(\mathbf{z}), \cdots, \pi_k(\mathbf{z}))$ and $\mathbf{u} \sim N(\cdot \,|\, \mathbf{0}, \mathbf{I}_d)$. Note that this transfer function is not differentiable w.r.t parameters $\{\pi_i(\mathbf{z})\}$ and the MDN weights predicting it, because of the indicator function. As such, it is not directly suitable for SGD (backpropagation) based end-to-end training of the autoencoder (see Algorithm 1). We next propose a differentiable approximation of the MDN transfer function based on the Gumbel softmax reparametrization (Jang et al., 2017), which is used in our autoencoder implementation.

## E.1  DIFFERENTIABLE MDN TRANSFER FUNCTION

Consider the transfer function of the MDN in Eq. (25). We would like to replace sampling from the categorical mixture prior $\mathrm{Cat}(\pi_1(\mathbf{z}), \cdots, \pi_k(\mathbf{z}))$ with a differentiable function that closely approximates it. We apply the *Gumbel-Softmax* reparametrization (Jang et al., 2017) which solves this exact problem. Recall that the component prior probabilities can be written in terms of the prior logits as:

$$\pi_i(\mathbf{z}) \;=\; \frac{e^{\alpha_i(\mathbf{z})}}{\sum_{j=1}^{k} e^{\alpha_j(\mathbf{z})}}, \quad \forall i \in [k].$$

Consider $k$ iid standard Gumbel random variables $G_1, \cdots, G_k \overset{\mathrm{iid}}{\sim} \mathrm{Gumbel}(0, 1)$. It can be shown that, for any $\mathbf{z} \in \mathcal{Z}$, the random variable

$$S(\mathbf{z}) \;=\; \underset{i \in [k]}{\mathrm{argmax}}\ G_i \;+\; \alpha_i(\mathbf{z}) \tag{26}$$

---

[10]This is the same as the prior distribution over the input messages.

follows the categorical distribution $\text{Cat}(\pi_1(\mathbf{z}), \cdots, \pi_k(\mathbf{z}))$. This standard result is known as the Gumbel-max transformation. While Eq. (26) can be directly used inside the indicator function in Eq. (25), the $\text{argmax}$ will still result in the transfer function being non-differentiable. Therefore, we use the following *temperature-scaled softmax* function as a smooth approximation of the $\text{argmax}$

$$\widehat{S}_i(\mathbf{z}\,;\tau) \;=\; \frac{\exp[(G_i + \alpha_i(\mathbf{z}))/\tau]}{\sum_{j=1}^{k} \exp[(G_j + \alpha_j(\mathbf{z}))/\tau]}, \;\; \forall i \in [k], \tag{27}$$

where $\tau > 0$ is a temperature constant. For small values of $\tau$, the temperature-scaled softmax will closely approximate the $\text{argmax}$, and the vector $[\widehat{S}_1(\mathbf{z}\,;\tau), \cdots, \widehat{S}_k(\mathbf{z}\,;\tau)]$ will closely approximate the one-hot vector $[\mathbb{1}(S(\mathbf{z}) = 1), \cdots, \mathbb{1}(S(\mathbf{z}) = k)]$.

Applying this Gumbel softmax reparametrization in Eq. (25), we define a modified differentiable transfer function for the Gaussian MDN as

$$\mathbf{x} \;=\; \widehat{\mathbf{h}}_{\boldsymbol{\theta}_c}(\mathbf{z}, \mathbf{u}) \;=\; \sum_{i=1}^{k} \widehat{S}_i(\mathbf{z}\,;\tau)\,(\boldsymbol{\sigma}_i^2(\mathbf{z}) \odot \mathbf{u} + \boldsymbol{\mu}_i(\mathbf{z})). \tag{28}$$

With this transfer function, it is straightforward to compute gradients with respect to the prior logits $\alpha_i(\mathbf{z}), \; \forall i$. Another neat outcome of this approach is that the stochastic components (Gumbel random variables $G_i$) are fully decoupled from the deterministic parameters $\alpha_i(\mathbf{z})$ in the gradient calculations with respect to $\widehat{S}_i(\mathbf{z}\,;\tau)$. In our experiments, we used this Gumbel-softmax based smooth transfer function while training the autoencoder, but during prediction (inference), we use the exact $\text{argmax}$ based transfer function. We found $\tau = 0.01$ to be a good choice for all the experiments.

## F  SIMULATED CHANNEL VARIATION MODELS

We provide details of the mathematical models used to create the simulated channel variations in our experiments. These models are frequently used in the study of wireless channels (Goldsmith, 2005).

### F.1  ADDITIVE WHITE GAUSSIAN NOISE (AWGN) MODEL

This is the simplest type of channel model where the channel output $\mathbf{x}$ is obtained by adding random Gaussian noise $\mathbf{n}$ to the channel input $\mathbf{z}$, *i.e.*, $\mathbf{x} = \mathbf{z} + \mathbf{n}$. It is assumed that the noise $\mathbf{n} \sim \mathcal{N}(\cdot \,|\, \mathbf{0}, \sigma_0^2\,\mathbf{I}_d)$ is independent of $\mathbf{z}$. We find the signal-to-noise ratio (SNR) of this channel, and specify how to set the noise variance $\sigma_0^2$ in order to achieve a target SNR value.

The average power in the signal component of $\mathbf{x}$ is given by $\mathbb{E}[\|\mathbf{z}\|_2^2] = p_{\text{avg}}$. The noise power in this case is given by $\mathbb{E}[\|\mathbf{n}\|_2^2] = \sigma_0^2$. The signal-to-noise ratio (SNR) for the AWGN model is therefore given by

$$\frac{E_b}{N_0} \;=\; \frac{\mathbb{E}\big[\|\mathbf{z}\|_2^2\big]}{2\,R\,\mathbb{E}\big[\|\mathbf{n}\|_2^2\big]} \;=\; \frac{p_{\text{avg}}}{2\,R\,\sigma_0^2},$$

where $R$ is the communication rate of the system in bits/channel use (O'Shea & Hoydis, 2017). To simulate an AWGN channel with a target SNR of $E_b/N_0$, we select the noise variance as follows:

$$\sigma_0 \;=\; \sqrt{\frac{p_{\text{avg}}}{2\,R\,(E_b/N_0)}}. \tag{29}$$

### F.2  UNIFORM FADING MODEL

The channel output $\mathbf{x} \in \mathbb{R}^d$ for this model as a function of the channel input (symbol vector) $\mathbf{z} \in \mathbb{R}^d$ is given by

$$\mathbf{x} \;=\; A\,\mathbf{z} + \mathbf{n},$$

where $A \sim \text{Unif}[0, a]$ is a uniformly-distributed scale factor, and $\mathbf{n} \sim \mathcal{N}(\cdot \,|\, \mathbf{0}, \sigma_0^2\,\mathbf{I}_d)$ is an additive Gaussian noise vector. Both $A$ and $\mathbf{n}$ are assumed to be independent of each other and $\mathbf{z}$. The average power in the signal component of $\mathbf{x}$ is given by

$$\widetilde{p}_{\text{avg}} \;:=\; \mathbb{E}\big[\|A\,\mathbf{z}\|_2^2\big] \;=\; \mathbb{E}\big[A^2\big]\,\mathbb{E}\big[\|\mathbf{z}\|_2^2\big]$$
$$=\; \frac{a^2}{3}\,p_{\text{avg}},$$

where $p_{\text{avg}}$ denotes the average power in the channel input $\mathbf{z}$. The noise power in this case is given by $\mathbb{E}[\|\mathbf{n}\|_2^2] = \sigma_0^2$. The signal-to-noise ratio (SNR) for this model is therefore given by

$$\frac{E_b}{N_0} = \frac{\mathbb{E}[\|A\,\mathbf{z}\|_2^2]}{2\,R\,\mathbb{E}[\|\mathbf{n}\|_2^2]} = \frac{a^2\,p_{\text{avg}}}{6\,R\,\sigma_0^2},$$

where $R$ is the communication rate of the system in bits/channel use. We select the fading factor $a$ such that the channel output has a target SNR value using the following equation:

$$a = \sqrt{\frac{6\,R\,\sigma_0^2\,(E_b/N_0)}{p_{\text{avg}}}}. \tag{30}$$

## F.3 RICEAN AND RAYLEIGH FADING MODELS

The channel output for the Ricean fading model is given by

$$\mathbf{x} = \mathbf{A}\,\mathbf{z} + \mathbf{n},$$

where $\mathbf{A}$ is a diagonal matrix with the diagonal elements $a_1, \cdots, a_d \overset{\text{iid}}{\sim} \text{Rice}(\cdot\,|\,\nu, \sigma_a^2)$ following a Rice distribution, and $\mathbf{n} \sim \mathcal{N}(\cdot\,|\,\mathbf{0}, \sigma_0^2\,\mathbf{I}_d)$ is an additive Gaussian noise vector. It is assumed that $\mathbf{n}$ and $\mathbf{A}$ are independent of each other and of $\mathbf{z}$. Note that Rayleigh fading is a special case of Ricean fading when the parameter $\nu = 0$. For this model, the average power in the signal component of $\mathbf{x}$ is given by

$$\widetilde{p}_{\text{avg}} := \mathbb{E}[\|\mathbf{A}\,\mathbf{z}\|_2^2] = \sum_{i=1}^d \mathbb{E}[a_i^2\,z_i^2] = \sum_{i=1}^d \mathbb{E}[a_i^2]\,\mathbb{E}[z_i^2]$$
$$= (2\,\sigma_a^2 + \nu^2)\,\mathbb{E}[\|\mathbf{z}\|_2^2] = (2\,\sigma_a^2 + \nu^2)\,p_{\text{avg}},$$

where $p_{\text{avg}}$ denotes the average power in the channel input $\mathbf{z}$. We used the fact that the second moment of the Rice distribution is given by $\mathbb{E}[a_i^2] = 2\,\sigma_a^2 + \nu^2$. It is useful to consider the derived parameters $K = \nu^2/2\,\sigma_a^2$ which corresponds to the ratio of power along the line-of-sight (LoS) path to the power along the remaining paths, and $\Omega = 2\,\sigma_a^2 + \nu^2$ which corresponds to the total power received along all the paths. The SNR for this model is given by

$$\frac{E_b}{N_0} = \frac{\mathbb{E}[\|\mathbf{A}\,\mathbf{z}\|_2^2]}{2\,R\,\mathbb{E}[\|\mathbf{n}\|_2^2]} = \frac{(2\,\sigma_a^2 + \nu^2)\,p_{\text{avg}}}{2\,R\,\sigma_0^2}.$$

For a given input average power and target SNR, the parameters of the Rice distribution can be set using the equation

$$2\,\sigma_a^2 + \nu^2 = \frac{2\,R\,\sigma_0^2\,(E_b/N_0)}{p_{\text{avg}}}.$$

To create channel variations of different SNR, we fix the variance $\sigma_a^2$ and vary the power of the LoS component $\nu^2$. Suppose the smallest SNR value considered is $S_{\text{min}}$, we set $\sigma_a^2$ using

$$2\,\sigma_a^2 = \frac{2\,R\,\sigma_0^2\,S_{\text{min}}}{p_{\text{avg}}}, \tag{31}$$

and set $\nu$ to achieve a target SNR $E_b/N_0$ using

$$\nu^2 = \frac{2\,R\,\sigma_0^2\,(E_b/N_0 - S_{\text{min}})}{p_{\text{avg}}}. \tag{32}$$

For this choice of parameters, the power ratio of LoS to non-LoS components is given by

$$K = \frac{E_b/N_0}{S_{\text{min}}} - 1.$$

The $K$-factor for Rician fading in indoor channel environments with an unobstructed line-of-sight is typically in the range 4 dB to 14 dB (Linnartz, 2001). Rayleigh fading is obtained for $K = 0$ (or $\nu = 0$).

Finally, note that the vector $\mathbf{z}$ is composed of one or more pairs of in-phase and quadrature (IQ) components of the encoded signal (*i.e.*, dimension $d = 2p$). Since each IQ component is transmitted as a single RF signal, the Ricean amplitude scale is kept the same for successive pairs of IQ components in $\mathbf{z}$. In other words, the amplitude scales are chosen to be $a_1, a_1, \cdots, a_p, a_p$. This does not change any of the above results.

