# OpenReview forum: "Few-Shot Domain Adaptation For End-to-End Communication"
_ICLR.cc/2023/Conference — ICLR 2023 notable top 25%_

### Official Review · Reviewer_jGKn · 2022-10-23

**Confidence:** 3
**Correctness:** 4
**Technical Novelty And Significance:** 3
**Empirical Novelty And Significance:** 3
**Recommendation:** 8

**Clarity, Quality, Novelty And Reproducibility:**

The paper is well organized and clearly stated. The advantage of the method is well stated.

**Strength And Weaknesses:**

Strength
1.	The paper is well organized and has rich details.
2.	The advantage of the proposed method is clearly stated and demonstrated.
3.	The adaptation approach is based on appropriate assumptions and is well supported by the properties of Gaussian mixtures.
4.	The effectiveness of the method is evaluated by both simulated and real experiments, and there are also experiments when the assumptions could not hold.

Weaknesses
1.	Some confusions.
In Parameter Transformation part, you state that “The number of adaptation parameters is given by k (2 d2 + d + 2). This is typically much smaller than the number of MDN parameters (weights and biases from all layers)”.
In previous part you state that “The MDN output with all the mixture parameters has dimension p = k (d(d + 1)/2 + d + 1).”
Why the adaptation parameters is much smaller than the number of MDN parameters?
2.	Some figures are not self-explanatory. For instance, in Figure 4, the line of No adapt or Finetune are covered by other lines, without additional explanation.
3.	More experiments. How the unsupervised domain adaptation performs based on the baseline model and how it compares with the proposed approach?


**Summary Of The Paper:**

In this paper, the authors provide a few-shot domain adaptation method to address the channel distribution changes of communication systems. Specifically, using the properties of Gaussian mixtures, they propose a solid domain adaption process for the generative channel model (MDN). Besides, they propose a input-transformation method which transform the input of decoder from target domain into source domain, without modifying the encoder-decoder networks. They also derive experiments on a mmWave FPGA platform and show the strong performance improvements of the proposed method.

**Summary Of The Review:**

The paper is well organized and has rich details. The work is based on appropriate assumptions and the properties of Gaussian mixtures, and the effectiveness is demonstrated by experiments on both simulated and real experiments.

---

> ### Author Response · Authors · 2022-11-14
> **Response to Reviewer jGKn**
>
> We thank the reviewer for their positive feedback and comments on our submission. Please find our responses below.
>
> ### Question on the number of adaptation parameters
> In section 2 (last line of page 3), we state that the MDN output, which predicts all the mixture parameters, has dimension $p = k (d(d + 1)/2 + d + 1)$. This includes the covariance matrix, mean vector, and component prior for the $k$ components of the Gaussian mixture. Please note that this is different from the number of parameters (weights) of the MDN network itself. For the MDN architecture we used in the experiments (see Table 3 in Appendix C.1), the number of parameters would be 12925.
>
> In Section 3.1 (under `Parameter Transformations`), we define the number of adaptation parameters to be $|\mathbf{\psi}| = k (2 d^2 + d + 2)$. This is also different from the size of the MDN output defined earlier. Taking specific values of $d = 2$ and $k = 5$, we compare these sizes below:
> ```
> Size of the MDN output: 30
> Number of adaptation parameters: 60
> Number of MDN parameters: 12925
> ```
> For this case (and more generally), the number of MDN parameters is much larger than the number of adaptation parameters.
>
> ### Clarity of the figures
> We agree with the reviewer that the performance curves of some baseline methods are overlapping in the figures, which makes it a bit unclear without explanation. Please note that we briefly discuss this in the last four lines of Section 4.1.
>
> We found the performance of the methods `Finetune` and `Finetune last` to be very close to that of `No adapt` in a majority of the experiments. As a result, the corresponding curves could not be clearly discerned in the plots. We have performed checks to make sure that this is not due to a bug or insufficient optimization (e.g., checking if the final weights of the MDN and decoder are different for both methods). For both methods, we tried a range of learning rates for the Adam optimizer and increased the number of epochs to a large number. We have reported the best-case results for these methods, which suggests that they are not effective at adaptation using small target domain datasets. We will explain this more clearly in the revised paper.
>
> ### Unsupervised domain adaptation
> We have briefly discussed the challenge of utilizing unsupervised domain adaptation (UDA) methods for this problem in the Related Work subsection (please see the last 3 lines of the first paragraph). For instance, adversarial DA methods are not suitable for this problem, which requires fast and frequent test-time DA, because of their high computational and sample complexity and the imbalance in the number of source and target domain samples.
>
> UDA methods such as DANN (Ganin et al., 2016) are designed for the setting where the target domain has *only* a large unlabeled dataset. Moreover, the target domain dataset is assumed to be available at training time, and the classifier is trained using both the labeled source and unlabeled target domain datasets. Another challenge of applying UDA methods to this problem is that it does not provide a way to adapt the generative channel model (MDN in this case). This is an important requirement since we need the channel model to track the changes in the channel condition.

---

### Official Review · Reviewer_91WK · 2022-10-25

**Confidence:** 4
**Correctness:** 3
**Technical Novelty And Significance:** 3
**Empirical Novelty And Significance:** 3
**Recommendation:** 6

**Clarity, Quality, Novelty And Reproducibility:**

These are all seem good. The presentation is clear, and the writing quality is above the bar. For the communication field, I think the novelty is sufficient. They also provide the source code in supplementary materials.

**Strength And Weaknesses:**

Pros:

1. This paper considers the frequent change of channel in communication systems. They treat the change of channel as distribution shift and link this practical problem with few-shot domain adaptation. I think this is novel and advanced enough in the field of communication.

2. The proposed solution is easy to follow and can be used in more general few-shot DA scenarios. In addition, this method has better real-time performance compared with previous works.

Cons:

1. This paper lacks the few-shot domain adaptation methods as baselines, e.g., [1]. Current baselines are all the basic FDA solutions, and I worry about their competitiveness.

2. The evaluation metric they use is only the SER. As the application research article, the performance of the proposed method in practical communication problems is essential. They need to show the advantages of their learning-based method over conventional methods.

3. I find that the number of target data per class is more than 10 in this paper, and maybe such an amount is beyond the scale of few-shot learning. Additional experiments with less than 7 samples per class are important.

[1] Mottian et al. Few-Shot Adversarial Domain Adaptation. NeurIPS, 2017.


**Summary Of The Paper:**

This paper models the changes of channel in a communication system as a few-shot domain adaptation problem. They employ the Gaussian mixture density network to specifically model the channel and propose a transformation to compensate for changes in the channel distribution. They perform experiments both on simulated channel distributions and FPGA.

**Summary Of The Review:**

This paper aims to address the varying channels in communication with the help of few-shot DA. Their solution is simple and easy to follow, and the corresponding theoretical and empirical analysis are thorough. However, some experimental settings (see above) are need to be further optimized.

---

> ### Author Response · Authors · 2022-11-18
> **Response to reviewer 91WK**
>
> We thank the reviewer for the positive and insightful feedback. Please find our responses below.
>
> ### Fewer target domain samples
> Please note that we report the performance for 5 samples per class (less than 10) in Figures 4, 5, and 6. Also, for the generative adaptation of the MDN channel alone (Appendix C.6 and Figure 13), we report the conditional log-likelihood for `2, 4, 6, 8,...` samples per class.
> In the communication setting, the adaptation data is typically obtained from the preambles of packets, and it may be reasonable to collect at least 5 - 10 samples per class before performing adaptation. However, we agree that it would be interesting to evaluate the performance with as low as 1 sample per class, as suggested in some prior works on few-shot domain adaptation.
>
> To understand this, we ran experiments on some simulated distribution changes reported in the paper with `1, 2, 4, 6 and 8` samples per class. Specifically, we report these *new results* in `Appendix C.7` of the revised paper for the following distribution changes: 1) Random Gaussian mixtures (Figure 14.a), and 2) Ricean fading to Uniform fading (Figure 14.b). We follow the same protocol as in the main paper and report the average results from multiple random trials.
>
> From Figure 14.b, corresponding to random Gaussian mixtures, we observe that the proposed method is able to improve performance (decrease SER) starting from 1 sample per class.
>
> However, in Figure 14.b (Ricean fading to Uniform fading), the proposed method increases the SER compared to no adaptation for 1 and 2 samples per class. The SER then starts to decrease significantly from 4 samples per class. We hypothesize that the reason for failure of the proposed method with 1 and 2 samples per class could be the strong distribution change and complexity of the target uniform fading channel. While this may seem concerning, please note that we chose these strong simulated distribution changes in order to demonstrate the potential improvements of our method. In practical wireless channels, the distribution changes are likely to be more gradual, so that the proposed method can usually adapt well with only a few samples per class.
>
> ### Few-shot domain adaptation baselines
> As discussed in the second paragraph of  the Related Work section, Motiian et al., 2017a focuses on the training-time few shot DA problem, where the goal is to learn a shared domain-invariant feature space (via an inference network) and a predictor network such that the classification accuracy on both the target and source domain data, mapped to the domain-invariant feature space, is high. Their method FADA has an adversarial DA formulation and learns a domain-class discriminator (DCD) to classify pairs of samples from the source or target domain into four groups based on their domain and class label pair. Following the adversarial training approach, it iteratively trains the DCD and the inference and predictor networks to maximize the domain-class confusion and minimize the expected classification loss.
>
> We face the following challenges in trying to adopt a method like FADA (Motiian et al., 2017a) to the end-to-end communication problem:
> - In our problem, *both* the generative channel model (here the MDN) and the decoder have to be adapted using the few-shot target data. The method FADA is applicable only for the classifier (here the decoder), but does not provide a way of adapting the MDN. The latter is important in order to keep tracking the changes in channel distribution.
> -  We focus on the *test-time* few-shot DA problem, where an existing classifier trained on labeled source domain data has to be adapted at test time to changing class-conditional input distributions (i.e. $p(\mathbf{x} | y)$). FADA requires both the source and target domain datasets at training time, and it can be computationally expensive to retrain for every new batch of target domain data (a key motivation for this work is to avoid frequent retraining).
>
> ### Other evaluation metrics
> In this paper, we focused on the classification and generative channel modeling aspect of the end-to-end communication problem. Therefore, our evaluation metric is the symbol error rate for most of the experiments, and conditional log-likelihood for the generative adaptation experiment (Appendix C.6). We agree with the reviewer that for a full evaluation of our method as part of a practical communication system, we should evaluate other metrics such as the throughput and adaptation speed relative to the coherence time of the channel. However, this would require a significant systems-level effort, which we leave as future work.

---

### Official Review · Reviewer_fua1 · 2022-10-27

**Confidence:** 3
**Correctness:** 3
**Technical Novelty And Significance:** 4
**Empirical Novelty And Significance:** 4
**Recommendation:** 8

**Clarity, Quality, Novelty And Reproducibility:**

**Clarity**: Good. It was generally easy reading the paper, thanks to really crisp text and a comprehensive background section. The minor issue I found is that some patterns in the results are not discussed (see concern 2, 3) The only nitpick I have are the figures (esp. Figures 4-6) where legends are highly illegible.

**Quality**: Good. While there are minor discrepancies the approach (e.g., performance slightly deteriorates when there is no distribution change, does not translate well to certain distribution changes), I think it can be overlooked in light of the remaining contributions.

**Novelty**: Very good. The authors tackle a very well motivated problem (see strength 2) and propose an insightful approach to tackle it (see strength 3).

**Reproducibility**: Very good. The main paper (esp. the large appendix) appears to contain many details of the approach. Additionally, the code is provided as well. I'm not sure if the authors plan to release the channels from the mmWave FPGA testbed.

**Strength And Weaknesses:**

### Strengths

**1. Extensive evaluation**
- The approach is evaluated rigorously with well-suited baselines and a range of scenarios (e.g., multiple types of domain shifts, real-world evaluation). I especially appreciate evaluations studying when the (reasonable) assumptions are violated.

**2. Motivation and relevant problem**
- While there has been a lot of attention on generative channel modelling recently, most works in my knowledge largely (and somewhat incorrectly) assume a stationary distribution. This paper takes a step in the right direction by addressing this pain-point.

**3. Insightful approach**
- The approach overall is insightful and makes sense. By learning an adapter network and learning parameters relevant for the domain shifts (e.g., like FiLM modules), it makes few-shot domain-adaptation more tractable.
- Furthermore, I find the choice of the channel model representation (MDNs) to also be sufficiently appropriate for the task (as opposed to GANs) for this study.

### Concerns

**1. "labeled set obtained for free"**
- The paper at multiple times claims that few-shot learning is especially possible since we can get labeled dataset for free -- I find this slightly confusing.
- Wouldn't the labeled dataset be split between the encoder (transmitter) and decoder (receiver) devices? As a result, for a party to have the full labeled dataset, isn't a prerequisite communicating labels back to the other party?

**2. Evaluation: Some observations unclear**
- I found some patterns in the evaluation was somewhat unclear and would appreciate the authors' answers on the questions below:
- (a) Oracle-approach gap in Figure 4/5: I'm slightly surprised that proposed approach's symbol error rate does not converge to the oracle with a reasonable number of additional examples (50 * 16-QAM classes = 800), given that there are 50 learnable parameters. Are the authors aware if convergence is possible with even higher examples? Morevover, what is the size of the source dataset?
- (b) Unchanged error rates in Figure 4/5 for many baselines: Are the authors aware of why the error rates of many baselines do not improve at all in spite of more training examples? Were the "finetune" baselines finedtuned only on the new data or a combination? In the case of combination, are domain-invariant features learnt?
- (nitpick) Please summarize the performance degradation discussions in Ricean fading experiments in the main paper.

**3. Evaluation: Performance under no distribution change**
- I appreciate that the authors also evaluate under a non-domain shifted dataset in Figure 10. Can the authors clarify why results drop in performance when there is no distribution change?
- Specifically, it appears that the adapter layers' parameters are initialized such that it produces a identity mapping (page 18), so I'm surprised that this nonetheless degrades performance.

**4. SNR=14-20 dB**
- Can the authors comment whether a SNR of 14-20dB (which to me appears really large) is a reasonable setting? Did the authors also evaluate SNR vs. error rates for the approach and baselines? I wonder if the results shown here apply only in high SNR regimes.

**Summary Of The Paper:**

- The paper addresses the problem of handling domain-shifts that arises in generative learnt channel models in E2E communication systems in a few-shot setting.
- The proposed domain adaptation approach is tailored around a Mixture Density Network (MDN) representing the channel model. In here, the approach:
    - learns an adapter layer, which models an affine transform of the original conditional channel distribution
    - introduces an additional regularization objective to ensure the adapter doesn't converge to bad/degenerate solutions
    - presents a feature transformation formulation on the decoder side to aid learning on the domain-shifted distributions
- The approach is evaluated extensively, covering multiple types of distribution changes in both synthethic settings as well on a high-resolution mmWave testbed.

**Summary Of The Review:**

The paper tackles a relevant bottleneck in generative channel modelling for E2E communication systems (i.e., they are trained assuming a stationary distribution, but this isn't the typical case). The approach is novel and intuitive in my opinion, and is further evaluated extensively in both simulated and real conditions. While I have some minor concerns (e.g., can one really have a labelled dataset for this task in practise?), I don't think they significantly affect the paper's claims and contributions.

---

> ### Author Response · Authors · 2022-11-15
> **Response to Reviewer fua1 (part 1)**
>
> We thank the reviewer for their positive feedback and detailed review. Please find our responses to your questions and concerns below.
>
> ### Labels and Labeled data (concern 1)
> We would like to clarify that the statement “class labels are available for free” is made in order to highlight the fact that class labels are easy to obtain in this end-to-end communication setting, unlike other domains (e.g. computer vision) where labeling data could be expensive. Since the transmitted message is also the class label, it is always available without additional effort during the data collection (from the packet preambles). However, please note that it is still challenging / expensive to collect a large number of samples for domain adaptation, as discussed in Section 1. In contrast, it may be easy to obtain plenty of unlabeled data in other domains such as computer vision, where labeling is expensive.
>
> In communication protocols, preambles are attached to the front of the packets for synchronization, carrier frequency offset correction, and other tasks. The preambles consist of sequences of known symbols (which have a one-to-one mapping to the messages). Therefore, these sequences can be used as the labeled dataset since the receiver obtains the distorted symbol and knows the ground truth.
> The proposed MDN adaptation and input transformation at the decoder do not incur any modifications to the encoder (transmitter side). The constellation learned by the autoencoder is kept fixed during adaptation. Therefore, using the preambles from a small number of packets, our method performs adaptation at the receiver side and maintains the symbol error rate performance without communicating any information back to the encoder.
>
> Please let us know if this needs further clarification. We will include a brief discussion on this in the revised paper.
>
> ### Oracle-approach gap (concern 2.a)
> The reviewer raises an interesting point. Does the performance of the proposed method approach that of the oracle retraining method as the size of the adaptation dataset increases? We note that the oracle method `Retrained autoenc` retrains the channel model, encoder and decoder networks jointly using Algorithm 1 in Appendix D.3. In our experiments, we used `20,000` samples from the target (domain) distribution for training the MDN channel in between each epoch of the encoder/decoder update, and used `300,000` samples for training the encoder/decoder networks with the MDN parameters fixed (the same dataset size was also used for training on the source domain). The retrained autoencoder therefore learns a custom *optimal* constellation (encoding) for the target distribution. Our adaptation method, on the other hand, does not modify the constellation learned by the autoencoder from the source domain data, which could be sub-optimal for the target domain. This restriction of not modifying the constellation is practically advantageous since there is no need to communicate the new symbols back to the transmitter. Also, there is no need to change the transmitter side encoding frequently. Since the proposed method does not have the full flexibility of the oracle retrained autoencoder, we do not expect its performance (SER) to converge to that of the latter under significant distribution change.
>
> For instance, Figure 12 (b) in the paper shows the learned autoencoder constellation for a Ricean fading channel of 20dB SNR. This is quite different from the optimal constellation for an AWGN channel of 14dB SNR (which is closer to a 16-QAM constellation).
>
> To understand this better, we have run some experiments to evaluate our method with a large adaptation dataset, and compare it to the oracle retrained autoencoder. In the table below, we report the SER performance of our method with 1000 samples per class (so 16,000 samples overall) on a subset of simulated distribution changes reported in the paper. We follow the same protocol as the main paper, and report the average SER from multiple random trials on a large test set from the target distribution.
> | Source and target domains       | Proposed method (50 samples/class) | Proposed method (1000 samples/class) | Retrained autoencoder |
> |---------------------------------|--------------------------------------|----------------------------------------|-----------------------|
> | Random Gaussian mixtures        | 0.0766                               | 0.0695                                 | 0.0154                |
> | AWGN to Uniform fading          | 0.2370                               | 0.2138                                 | 0.1345                |
> | Ricean fading to Uniform fading | 0.1945                               | 0.1815                                 | 0.1345                |

---

> > ### Author Response · Authors · 2022-11-15
> > **Response to Reviewer fua1 (part 2)**
> >
> > #### Continuing the discussion on the oracle-approach gap:
> > We also report the performance of our method with 50 samples/class for comparison. From the table, we observe that there is a gap in the SER of our method (with 1000 samples/class) and that of the retrained autoencoder. We believe that allowing the encoder symbols to be optimized might be required in order to bridge this gap in SER. In conclusion, when it is possible to collect sufficient (large) data from the target distribution, it might be better to allow the encoder network to also be optimized via retraining. We will add this result and discussion to the revised paper in Appendix C.
> >
> > ### Unchanged Error Rates (concern 2.b)
> > We first provide some details on how the baselines `Finetune` and `Finetune last` are trained. They use only the target domain adaptation data (i.e. the new data), and do not use the source domain dataset. Both the methods first initialize the target domain MDN, encoder, and decoder networks with the corresponding parameters from the source domain. The method `Finetune` first finetunes *all* the MDN parameters to minimize the conditional log-likelihood of the target dataset using the Adam optimizer. Details on this are given in the third paragraph of Appendix C.1. After the MDN is finetuned, we freeze the parameters of the MDN and encoder and train only the decoder using data from the updated MDN channel. The method `Finetune last` differs from `Finetune` in that it optimizes only the weights of the final MDN layer.
> >
> > As pointed out by the reviewer, the baselines `Finetune` and `Finetune last` have very similar performance compared to the case of no adaptation. We have investigated this carefully and verified that this is not due to a bug or insufficient optimization (e.g., by checking if the final weights of the MDN and decoder are different for both methods). For both methods, we tried a range of learning rates for the Adam optimizer and increased the number of epochs to a large number (beyond 200 was not helpful). We have reported the best-case results for these methods, which suggests that they are not effective at adaptation using small target domain datasets. As mentioned in the last four lines of Section 4.1, we hypothesize that using the KL-divergence based regularization and constraining the number of adaptation parameters leads to more effective performance of our method.
> >
> > Since the fine-tuning baselines did not use a combination of the source and target domain datasets, we don’t expect them to learn domain-invariant representations.
> >
> > ### Evaluation: Performance under no distribution change
> > We agree with the reviewer that ideally, there should not be any degradation in the performance when there is no distribution change. We note that in Figure 10, the increase in SER of our method is quite small. For instance, with 5 samples per class, it increases to 0.018 from 0.015 for the Rician fading (14dB) channel, and from 0.24 to 0.25 for the Uniform fading (14dB) channel.
> > One way to mitigate this would be to increase the regularization constant $\lambda$, which would constrain the KL-divergence and prevent large changes in the adaptation parameters. Since we set $\lambda$ automatically using the validation metric, it is possible that the choice of $\lambda$ does not correspond to the best solution (in this case).
> >
> > As mentioned in the paper (2nd paragraph under `Performance Under No Distribution Change` in Appendix C.4), a practical system that frequently adapts to changes in the channel distribution should first have a distribution change-detection algorithm that takes a batch of new samples from the channel, and tests whether there is any change in the distribution or a drop in the autoencoder performance prior to adaptation. These measures can be used to better handle the scenario of no distribution change.
> >
> > ### SNR=14-20 dB
> > Our process for selecting this range of SNR was by first evaluating the error rate vs. SNR curve of the autoencoder for the different simulated channel distributions. We found that going below 14dB SNR results in a degradation of the autoencoder’s performance  (except perhaps for the AWGN channel, which we don’t use as a target distribution). Also, going above 20dB SNR did not lead to a significant decrease in the error rate. For the channels such as Ricean fading and Uniform fading, we found that even a retrained autoencoder has a relatively high error rate for lower SNRs.
> >
> > We thank the reviewer for bringing up these insights, which will help us improve the quality of the paper. We will submit a revised paper before the rebuttal deadline.

---

### Author Response · Authors · 2022-11-19
**Summary of changes to revised paper**

We thank all the reviewers for their constructive feedback that helped us improve our submission. We have revised the paper and added clarifications where needed based on the reviews (summarized below):

- Added a discussion on labels and labeled data in the communication setting in Appendix B.1 (Reviewer `fua1`).
- Added details on the fine-tuning baseline methods, and a discussion of why they have very similar error rates in Appendix C.1 (Reviewers `fua1` and `jGKn`).
- Added a paragraph on the choice of SNR range of 14dB to 20dB in Appendix C.1 (Reviewer `fua1`).
- Added some results and discussion in Appendix C.7 (Figure 14) on the performance of our method with fewer than 5 samples per class (Reviewer `91WK`).
- Added some results and discussion in Appendix C.8 (Table 4) on the gap between the proposed method (with a large target dataset) and the oracle retrained autoencoder (Reviewer `fua1`).

The changes have been highlighted in blue font. When an entire sub-section has been added/modified, only the section heading is highlighted.

---

### Decision · Program_Chairs · 2023-01-20

**Decision:**

Accept: notable-top-25%

**Justification For Why Not Higher Score:**

it would be unfair to the authors of the paper to state an arbitrary reason for not attributing a higher score. The paper has a significant potential to be impactful as it addresses a practical problem and provide an efficient solution by tackling an existing problem in communication systems with realistic assumption.

- The paper received a very high grade (7.33) from reviewers with a decent confidence level (3.33) from reviewers. The following reason can be considered for not attributing the paper the mention of oral:
    - Not considering a full evaluation of the proposed solution which takes into account metrics reflecting its practical aspect (To be objective and fair, it is important to mention that this would require a significant systems-level effort, which can be left for future work as stated by the authors);
    - The method is presented as a solution to a communication system problem. From the title of the paper, one can think that it is targeting a narrower audience of researchers while the proposed method is generic (which is of benefit to a greater audience and should be considered as a main strong point).

Determining if a paper should receive an award or a mention in a venue according to some arbitrary criterion could be meaningless. In the history of machine learning, great impactful foundational works, such as Generative Adversarial Networks (Goodfellow et al, 2014), were not accepted as spotlights or orals and yet their long-term impact was the true reward that authors could reap.
Other works presenting and proposing a solution designed to a very specific problem have been adopted later by a more general audience of researchers and applied to a broader spectrum of applications. To name a few, the UNet architecture (Ronneberger et al, 2015) was originally designed for biomedical image segmentations and has been adopted largely in the computer vision and speech community for different tasks.

References:
- Goodfellow, Ian, et al. "Generative adversarial networks." (NeurIPS 2014).
- Ronneberger, Olaf, Philipp Fischer, and Thomas Brox. "U-net: Convolutional networks for biomedical image segmentation." International Conference on Medical image computing and computer-assisted intervention. Springer, Cham, 2015.


**Justification For Why Not Lower Score:**

Reviewers unanimously recommended acceptance of the paper and most reviewers agreed that the paper is marginally above the threshold. The paper has the potential to be impactful as it introduces the first differentiable neural ray tracer for wireless channel modelling and new datasets for benchmarking purposes are provided.

**Metareview: Summary, Strengths And Weaknesses:**

I Summary:

I.1 Investigated Problem:
- The authors investigate the frequent changes of channels in communication systems and treat these changes as shifts in distribution (domain). The framework of few-shot domain adaptation is then considered to tackle the problem in question.

I.2 Proposed Solution:
- Leveraging the properties of Gaussian mixtures, the proposed domain adaptation approach is tailored around a Mixture Density Network (MDN) representing the channel distribution.
- A regularized, parameter-efficient adaptation of the MDN using a set of affine transformations is proposed.
The learned affine transformations are then used to design an optimal transformation at the decoder input to compensate for the distribution shift, and effectively present to the decoder inputs close to the source distribution.

I.3 Validity Proof of the Proposed Solution:
- An extensive evaluation is conducted, covering multiple types of distribution changes in both synthetic settings as well on a high-resolution mmWave testbed.

II Strengths:

- II.1 From a structural point of view:
    - The paper is well-written. The method is clearly presented and ideas are easy to follow.

- II.2 From an analytical point of view:
    - The approach is:
        - novel;
        - evaluated rigorously;
        - compared with well-suited baselines;
        - considers a range of scenarios with different types of domain shifts, real-world as well as synthetic evaluations;
        - takes a step in the right direction by considering the more realistic assumption of the non-stationary distribution of a channel in communication systems.

- II.3 From a perspective of soundness (development, unity, and coherence) and completeness (correctness):
    -  The strength points mentioned above are sufficient evidence of the soundness and completeness of the paper. The transparency aspect of the submission is also appreciated as open-source code is provided for reproducibility purposes and other details are contained in the appendix.

III Addressing what can be thought of as weaknesses:

- The authors tried to address all the concerns raised by the reviewers (all of them were minor). It is worth mentioning that concern about the experimental setting was raised as a state-of-the-art few-shot adversarial domain adaptation method (FADA [1]) was not considered in the comparison conducted with existing baselines. The authors provide an explanation for the reason why FADA was not included in the comparison as it does not provide a way of adapting the mixture density network which is an important component to keep track of changes in channel distribution.

- Reviewers also mentioned the necessity of a full evaluation of the proposed solution that takes into account metrics reflecting the practical aspect of the method. The authors agreed that other metrics such as the throughput and adaptation speed relative to the coherence time of the channel could be considered as part of a practical communication system. Yet, this would require a significant systems-level effort, which can be left for future work.

IV Potential of the paper:

- IV.1 From a Potential perspective (Potential of the paper to the community): The proposed solution has great potential to be of benefit to the whole community, especially researchers interested application of deep learning in communication systems.

V References:

- Motiian, Saeid, et al. "Few-shot adversarial domain adaptation." Advances in neural information processing systems (2017).


**Note From Pc:**

if the above contains the word "oral" or "spotlight" please see: "oral" presentation means -> notable-top-5% and "spotlight" means -> notable-top-25%. As stated in our emails, we are disassociating presentation type from AC recommendations

**Summary Of Ac-Reviewer Meeting:**

N/A